

# Quantity and distribution of methane entrapped in sediments of calcareous, Alpine glacier forefields

Biqing Zhu[1], Manuel Kübler[1], Melanie Ridoli[1], Daniel Breitenstein[1], Martin H. Schroth[1]

[1]Institute of Biogeochemistry and Pollutant Dynamics, ETH Zurich, Zurich, CH-8092, Switzerland

*Correspondence to*: Martin H. Schroth (martin.schroth@env.ethz.ch)

**Abstract.** Aside from many well-known sources, the greenhouse gas methane ($CH_4$) was recently discovered entrapped in sediments of Swiss Alpine glacier forefields derived from calcareous bedrock. A first study performed in one glacial catchment indicated that $CH_4$ was ubiquitous in sediments and rocks, and was largely of thermogenic origin. Here we present results of

a follow-up study, which aimed at (1) determining occurrence and origin of sediment-entrapped $CH_4$ in other calcareous glacier forefields across Switzerland, and (2) providing an inventory for this sediment-entrapped $CH_4$, i.e., determining contents and total mass of $CH_4$ present, and its spatial distribution within and between five different Swiss glacier forefields situated on calcareous formations of the Helvetic Nappes of the Central Alps.

Sediment and bedrock samples were collected at high spatial resolution from the forefields of Im Griess, Griessfirn,

Griessen, Wildstrubel, and Tsanfleuron glaciers, representing different geographic and geologic regions of the Helvetic Nappes. We performed geochemical analyses on gas extracted from sediments and rocks, including determination of $CH_4$ contents, stable carbon-isotope analyses ($\delta^{13}C_{CH4}$), and determination of gas-wetness ratios (ratio of $CH_4$ to ethane and propane contents). To estimate the total mass of $CH_4$ entrapped in glacier-forefield sediments, the total volume of sediment was determined based on measured forefield area and either literature values of mean sediment depth or direct depth measurements

using electrical-resistivity tomography.

Methane was found in all sediments (0.08–73.81 µg $CH_4$ g$^{-1}$ dry weight) and most rocks (0.06–108.58 µg $CH_4$ g$^{-1}$) collected from the five glacier forefields, confirming that entrapped $CH_4$ is ubiquitous in these calcareous formations. Geochemical analyses further confirmed a thermogenic origin of the entrapped $CH_4$ (average $\delta^{13}C_{CH4}$ of sediment: -28.23 (± 3.42) ‰; average gas-wetness ratio: 75.2 (± 48.4)). Whereas sediment-entrapped $CH_4$ contents varied moderately within

individual forefields, we noted a large, significant difference in $CH_4$ content and total $CH_4$ mass (range: 200–3881 t $CH_4$) between glacier forefields at the regional scale. Lithology and tectonic setting within the Helvetic Nappes appeared to be dominant factors determining rock and sediment $CH_4$ contents. Overall, a substantial quantity of $CH_4$ was found to be entrapped in Swiss calcareous glacier forefields. Its potential release and subsequent fate in this environment is the subject of ongoing studies.



# 1 Introduction

The atmospheric concentration of the greenhouse gas methane ($CH_4$) has increased from pre-industrial values < 0.8 µL/L to a current global average of ~1.86 µL/L (Dlugokencky), indicating an imbalance in strength between $CH_4$ sources and sinks during this time period (Kirschke et al., 2013; Saunois et al., 2016; Ciais et al., 2013). Methane sources are commonly classified as either natural, or, in case they result from human activity, anthropogenic. Major natural $CH_4$ sources include wetlands, inland waters such as lakes, rivers, and reservoirs, as well as geological sources, e.g., gas seeps and geothermal/volcanic sources (Etiope et al., 2008; Kirschke et al., 2013; Bastviken et al., 2011). Major anthropogenic $CH_4$ sources include rice paddies (Saunois et al., 2016), livestock husbandry (Johnson et al., 2002), fossil fuels (Bousquet et al., 2006), and biomass burning (Bousquet et al., 2006; Kirschke et al., 2013). On the other hand, the major sink for atmospheric $CH_4$ is its chemical oxidation by hydroxyl radicals in the troposphere, accounting for up to 90 % of the global $CH_4$ sink (Kirschke et al., 2013; Bousquet et al., 2006). Aerated soils serve as an additional sink for atmospheric $CH_4$. They harbor a group of aerobic methane-oxidizing bacteria (MOB), which catalyze $CH_4$ oxidation at near-atmospheric concentrations to largely cover their carbon and energy needs (Curry, 2009; Zhuang et al., 2013; Dunfield, 2007). Although major $CH_4$ sources and sinks have been identified, a fair amount of uncertainty remains regarding their magnitude and thus their contribution to atmospheric $CH_4$ concentration (Wei et al., 2015; Spahni et al., 2011; Saunois et al., 2016).

An alternative way to categorize $CH_4$ sources is based upon the $CH_4$ production pathway: microbial, thermogenic, or abiotic (Conrad, 2009; Etiope and Sherwood Lollar, 2013; Joye, 2012; Whiticar, 1999). Microbial $CH_4$, which accounts for ~70 % of global $CH_4$ emissions to the atmosphere, is produced by methanogenic archaea (methanogens) under anoxic conditions and in the absence of energetically more favorable terminal electron acceptors as the final step of organic matter degradation (Conrad, 1996; Conrad, 2009; Denman et al., 2007). Conversely, thermogenic $CH_4$ is produced in sedimentary deposits under elevated temperatures and pressures during sediment diagenesis by thermal decomposition of organic matter (Etiope, 2012; Martini et al., 2003; Schoell, 1988). This type of $CH_4$ is dominant in natural-gas fossil fuel, and is often found in terrestrial and marine gas seeps and mud volcanoes (Etiope, 2012; Kirschke et al., 2013; Etiope, 2009). Together, microbial and thermogenic $CH_4$ are frequently referred to as biotic $CH_4$, as in both cases the initial substrates are of biological origin (Etiope and Sherwood Lollar, 2013). Finally, $CH_4$ can also be formed via inorganic chemical reactions in the Earth's crust and mantle, e.g., in serpentinized, ultramafic rocks, and is therefore referred to as abiotic $CH_4$ (Etiope and Sherwood Lollar, 2013; Etiope and Schoell, 2014; Etiope et al., 2018). Initial substrates of abiotic $CH_4$ production typically include CO, $CO_2$, and $H_2$ (Etiope and Schoell, 2014). Stable isotope analyses and/or analyses of gas composition are commonly employed to distinguish between microbial, thermogenic, and abiotic $CH_4$ origins (Etiope and Schoell, 2014; Whiticar, 1999; Milkov and Etiope, 2018; Schoell, 1988).

Accelerated melting of many glaciers and ice sheets as a result of global warming (Haeberli et al., 2007; Paul et al., 2004; UNEP and WGMS, 2008) has prompted glacial environments to become topics of intense research in recent years, including investigations on their role in the turnover of greenhouse gases. Specifically, several studies have identified





subglacial environments as habitats for methanogens, and consequently as a potentially important $CH_4$ source (Wadham et al., 2012; Wadham et al., 2013). Evidence thereof was provided from elevated $CH_4$ concentrations found in basal ice (Campen et al., 2003; Christner et al., 2012; Souchez et al., 1995), and from long-term incubation experiments that indicated a substantial methanogenic potential in subglacial sediments (Wadham et al., 2012; Stibal et al., 2012; Boyd et al., 2010). Emissions of $CH_4$ from subglacial environments to the atmosphere, in particular through meltwater discharge, have recently been confirmed and quantified in field measurements (Burns et al., 2018; Christiansen and Jørgensen, 2018; Lamarche-Gagnon et al., 2019).

An alternative potential $CH_4$ source in glacial environments was recently detected in sediments of Swiss glacier forefields, in particular in those derived from calcareous bedrock (Nauer et al., 2012). Subsequent laboratory experiments revealed that this $CH_4$ could be released from these sediments upon mechanical impact and during acidification (Nauer et al., 2014). In a recent study focusing on one particular Swiss glacial catchment (Wildstrubel catchment, Canton Valais), we established that entrapped $CH_4$ was virtually omnipresent in sediment and bedrock samples collected throughout this catchment, but that $CH_4$ contents exhibited substantial variation between sampling locations (Zhu et al., 2018). We also provided robust evidence based on stable-isotope and other geochemical data that $CH_4$ entrapped in sediment and bedrock samples was predominantly of thermogenic origin, and that microbial $CH_4$ production was likely of minor importance at this site. However, as the focus of that study was on the occurrence and origin of entrapped $CH_4$ in different regions of the catchment, the number of samples collected was insufficient to rigorously assess spatial distribution and total quantity (here defined in terms of content, i.e., concentration, and total mass) of entrapped $CH_4$ within the forefield sediments (Zhu et al., 2018). Yet, to better characterize this potential $CH_4$ source, it is important to assess its spatial distribution and total quantity, particularly in glacier-forefield sediments, as we expect the potential for $CH_4$ release from these sediments to far exceed that from large bedrock surfaces due to the much higher specific surface area of the former (André et al., 2009; Michel and Courard, 2014). Moreover, as calcareous glacier-forefield sediments throughout the Swiss Alps are of similar origin (Weissert and Stössel, 2015), sediment-entrapped $CH_4$ may be a feature common to most if not all Swiss glacier forefields derived from calcareous bedrock. Whereas this hypothesis remains to be tested, its confirmation would greatly increase the magnitude of this potential $CH_4$ source.

Therefore, the overall goal of this study was to extend the work of Zhu et al. (2018) to other calcareous glacier forefields located in different regions of the Swiss Alps, and to assess the distribution of entrapped $CH_4$ contents within and compare total mass of entrapped $CH_4$ between all sampled glacier forefields. Specific objectives included to (1) test occurrence and origin of sediment-entrapped $CH_4$ in four additional calcareous glacier forefields. Furthermore, we wanted to (2) assess the spatial distribution of sediment-entrapped $CH_4$ contents in detail within one glacier forefield, testing for dependencies on sediment depth, sediment age, and glacier-forefield landforms, and based on the results obtained to (3) efficiently sample sediments of the other glacier forefields to quantify contents and total mass of sediment-entrapped $CH_4$. Finally, we wanted to (4) upscale these results and derive a first estimate of the total mass of sediment-entrapped $CH_4$ contained in all Swiss glacier forefields situated on calcareous bedrock.



## 2 Methods

### 2.1 Field sites and field-work stages

Field work was conducted in five different glacier forefields: Im Griess (IMG), Griessfirn (GRF), and Griessen (GRI) glaciers located in Central Switzerland in Cantons Uri (IMG, GRF) and Obwalden (GRI), and Tsanfleuron (TSA) and the previously
investigated Wildstrubel (WIL, Zhu et al. (2018)) glaciers located in Canton Valais (Figs. 1 and S1). These forefields were selected for two main reasons. Foremost, their sediments are mainly derived from calcareous bedrocks of the Helvetic Nappes (green shaded area in Fig. 1), which consist of a series of nappes (sheets of thrusted rocks) largely composed of Mesozoic limestones, shales, and marls of Jurassic to Eocene age (Pfiffner, 2014; Weissert and Stössel, 2015). They were originally deposited on the shallow northern shelf of the ancient Alpine Tethys Ocean (Weissert & Mohr, 1996), and subsequently
deformed, folded, and stacked on top of each other during Alpine orogeny (Herwegh and Pfiffner, 2005). Whereas individual nappes within the Helvetic Nappe system therefore share a similar origin, lithology and tectonic settings between individual nappes can be quite diverse (Weissert and Stössel, 2015). This was suggested to be a dominant factor determining rock $CH_4$ contents in the WIL catchment (Zhu et al., 2018). Consequently, we chose to investigate distant glacier forefields within the Helvetic Nappes (e.g., distance TSA to IMG ~136 km), for which sediments are derived from different individual nappes, but
also glacier forefields in close proximity to each other (e.g., distance IMG to GRF ~3.8 km; TSA to WIL ~24 km), for which sediments are derived, at least in part, from the same nappe. A second important reason for selection was that all five glacier forefields are relatively easy to access, facilitating sample collection and transport to the laboratory.

We conducted our field work in two stages. During stage I in summer 2016, we performed a detailed investigation on the spatial distribution of sediment-entrapped $CH_4$ within a designated sampling zone at the GRF glacier forefield, using high
spatial-resolution sampling to determine variations in entrapped $CH_4$ contents in relation to sediment depth, sediment age, and glacier-forefield landforms. The GRF forefield was chosen for this purpose mainly because it features well-defined sediment-age classes and well-developed, clearly distinguishable landforms within a previously characterized sampling zone (Chiri et al., 2015; Chiri et al., 2017). We also conducted measurements of sediment thickness (distance between the ground surface and the underlying bedrock) to estimate sediment volumes and thus the total mass of entrapped $CH_4$ present in these sediments.
Results of the GRF field work were then used to adapt our sampling strategies for field-work stage II performed in summer 2017, to quantify contents and total mass of sediment-entrapped $CH_4$ in the IMG, GRI, TSA, and WIL glacier forefields. During both field-work stages, selected sediment and rock samples were used to identify the origin of the entrapped $CH_4$ based on $CH_4$ stable carbon-isotope analyses and analyses of entrapped gas composition (see below).

### 2.1.1 Field-work stage I (GRF glacier forefield)

Sampling and measurements during stage I in the GRF forefield was conducted in three steps. First we tested the effect of sediment depth, then the effects of sediment age and glacier-forefield landforms on entrapped $CH_4$ contents. Finally, we



estimated the total mass of sediment-entrapped $CH_4$ based on measured $CH_4$ content, sediment thickness, and sediment-covered area.

To study the effect of sediment depth on entrapped $CH_4$ contents, we implemented a completely randomized design,
selecting 14 random locations within our sampling zone (not shown). We collected a total of 52 sediment samples (each ~500 g) by excavation from depths ranging from 20 to 70 cm below ground surface. All sediment samples were stored in clean plastic bags, transferred to the laboratory, and kept in the dark at 4 °C before further treatment. Following the extraction of entrapped gas and subsequent quantification of $CH_4$ contents in sediment samples (see below), the effect of sediment depth on entrapped $CH_4$ contents was studied using a one-way ANOVA.

To study the effect of sediment age and glacier-forefield landforms on entrapped $CH_4$ contents, we implemented a randomized block-sampling design. We first divided the GRF sampling zone into nine blocks (a combination of three sediment-age classes and three landforms, Fig. 2a), adopting a previous classification (Chiri et al., 2017). The three sediment-age classes were: A (0–20 yr), B (20–50 yr) and C (50–100 yr). In this context, sediment age refers to the number of years since the sediment has been exposed to the atmosphere as a result of glacier retreat. The three forefield landforms at GRF were
floodplain, terrace, and sandhills. A floodplain refers to the frequently flooded area in the immediate vicinity of the glacial stream, which commonly consists of sediments of fine particle size (mostly silt) and a lack of vegetation. A terrace refers to an elevated, previously flooded area, i.e. a former floodplain, usually featuring some vegetation coverage. Finally, sandhills consist of un-oriented, hummocky glacial-debris deposits, typically featuring poorly sorted, well-aerated sediments of sandy loam to sandy clay-loam texture. We collected a total of 78 sediment samples (each ~500 g) by excavation from a depth of 20
cm below ground surface, with 8–12 samples collected at random locations from within each block (Fig. 2a). The sampling depth of 20 cm below ground surface was chosen based on our results from the previous step. Following laboratory analyses (see below), the impact of sediment age and landforms on entrapped $CH_4$ contents was studied using a two-way ANOVA.

In addition to sediments, we also collected a total of 17 bedrock samples from outcrops and large boulders within the GRF glacier forefield. These samples were used to determine the $CH_4$ content of the parent material (Zhu et al., 2018). All
bedrock samples were stored in plastic bags, transferred to the laboratory, and stored in dark at 4 °C before further treatment.

Estimation of the total mass of $CH_4$ entrapped in glacier-forefield sediments also requires information on sediment thickness. For the GRF sampling zone we employed the electrical resistivity tomography (ERT) method (e.g., Kneisel, 2006; Reynolds, 1997; Scapozza et al., 2011). Five two-dimensional, vertical ERT profiles (ERT1–ERT5) were measured during two field campaigns, covering the three sediment-age classes and the three landforms (Fig. 2b). Two profiles were measured
parallel to the glacier stream (ERT2 and ERT5), and three perpendicular to the glacier stream (ERT1, 3, and 4). For each profile, 48 stainless-steel electrodes (30 cm long, 1.2 cm dia.) were hammered into the sediment to a depth of ~15 cm and connected to two 24-core copper cables, which were linked to the ERT instrument (SYSCAL Pro; Iris Instruments, Orléans, France) at the profile's midpoint. To improve electrical coupling of the electrodes with the skeleton-rich glacier-forefield sediments, water-soaked sponges were positioned at the sediment surface surrounding each electrode. Profile ERT1 was
measured with an electrode interspacing of 2.5 m (total profile length 120 m), the other four with 5 m distance between the



electrodes (240 m profile length). Using a so-called Wenner-Schlumberger configuration (Loke, 2001), an electrical current was sent to the subsurface using a pair of electrodes. The voltage difference measured across the other pairs of electrodes was used to calculate the electrical resistivity of the subsurface. To infer the location of the sediment-bedrock interface, inversion of apparent resistivities was performed using the 2-D program RES2DINV (Loke and Barker, 1996). The average sediment

thickness and its uncertainty within the GRF forefield was then analyzed in R. Electrical resistivities >2000 $\Omega$m were considered indicative of solid bedrock, whereas resistivities < 2000 $\Omega$m were considered indicative of unconsolidated sediment (Kneisel, 2006; Reynolds, 1997). Portions of the ERT profiles, for which the sediment-bedrock interface could not be detected, were omitted from further analyses.

### 2.1.2 Field-work stage II (IMG, GRI, WIL, and TSA glacier forefields)

During stage II, we collected a total of 111 sediment samples at 20 cm depth, 25 samples from IMG, 25 from GRI, 33 from WIL, and 28 from TSA glacier forefields (sampling locations shown in Fig. 3). Based on results obtained during field-work stage I, and given that glacier-forefield landforms were much less prominent at IMG, GRI, WIL, and TSA, we divided each of the four forefields into six blocks, and collected four to eight sediment samples (each ~500 g) from each block at random locations. We also collected 55 bedrock samples from outcrops and boulders; 13 from IMG, 14 from GRI, 12 from WIL, and

16 from TSA glacier forefields (locations also shown in Fig. 3).

## 2.2 Laboratory procedures

### 2.2.1 Extraction of entrapped gas

We extracted entrapped gas from sediments and rocks using the acidification method described in Nauer et al. (2014) and Zhu et al. (2018). Before acid treatment, sediments were sieved with a clean 20 mm mesh sieve. Particles >20 mm were excluded

from subsequent analyses. For each sample, ~3–5 g of sediment was weighed and transferred into a 117 mL serum bottle, sealed with a butyl rubber stopper and crimped with an aluminum cap. The vial's headspace was then flushed with $N_2$ gas. Thereafter, 5 mL deionized water was added into the vial, followed by ~50 mL of 6 N HCl to dissolve carbonate minerals. The headspace of each vial was connected to one or multiple 1 L gas bags (Tesseraux GmbH, Bürstadt, Germany). Sediment samples released large amounts of gas immediately after the acid was added. When bubbling stopped, an additional 2 mL 6N

HCl was added to each vial to confirm that the carbonate minerals were fully dissolved. Full dissolution of all carbonate minerals took ~4 h. After gas extraction, ~200 mL of gas were removed from the gasbags with syringes and stored in glass vials for further analysis. The total volume of gas remaining in gas bags was measured with a mass-flow meter (Bronkhorst, Reinach, Switzerland). Rocks were first hammered or sawed into ~1 cm diameter pieces and then dissolved in the same way as sediments. Initial tests indicated that rock hammering or sawing had no adverse effect on measured entrapped $CH_4$ contents,

nor on other geochemical parameters.





### 2.2.2 Quantification of methane, ethane, and propane

Concentrations of CH$_4$ were measured with a gas chromatograph equipped with a flame-ionization detector (GC-FID; Trace GC Ultra, Thermo Electron, Rodano, Italy) and a Porapak N100/120 column. The column-oven temperature was 30 °C, runtime was 36 s. Nitrogen carrier-gas flow was set to 26 mL/min. The FID was operated at 150 °C in high sensitivity mode. Concentrations of ethane (C$_2$H$_6$) and propane (C$_3$H$_8$) were quantified in selected gas samples using the same GC-FID system, but with oven temperature at 40 °C for 2 min, an increase to 140 °C at a rate of 25 °C/min, and constant oven temperature of 140 °C for another 9 min. Gas contents were calculated as the mass of CH$_4$, C$_2$H$_6$, and C$_3$H$_8$ released during acidification, normalized to the dry weight of the sample. The dry weight of sediments was determined by oven-drying of subsamples at 60 °C for 72 h. Computed entrapped gas contents $C_{CH4}$, $C_{C2H6}$, and $C_{C3H8}$ were subsequently used to calculate the gas-wetness ratio as $C_{CH4}/(C_{C2H6} + C_{C3H8})$ (Jackson et al., 2013), a commonly used indicator of CH$_4$ origin (a value >1,000 is considered evidence for microbial CH$_4$, whereas a value <<1,000 is considered indicative of thermogenic CH$_4$ (Rowe and Muehlenbachs, 1999)).

### 2.2.3 Stable carbon-isotope analysis of entrapped methane

About five sediment samples and five bedrock samples from each glacier forefield were selected for stable carbon-isotope analysis of entrapped CH$_4$ ($\delta^{13}C_{CH4}$). To determine $\delta^{13}C_{CH4}$ we used a modified acidification protocol for gas extraction, which consisted of flushing the vials' headspace with He instead of N$_2$ to remove ambient air. Gas released during the acidification treatment was passed through two 1 M NaOH solutions to remove the majority of CO$_2$, an Ascarite trap to remove final traces of CO$_2$, a Drierite trap to remove H$_2$O vapor, and a 1 M ZnCl$_2$ trap to remove potential H$_2$S (all chemicals from Sigma Aldrich, Buchs, Switzerland). The purified gas samples were subsequently analyzed by GC-IRMS (Isoprime, Elementar Ltd., Stockport, UK).

### 2.3 Estimation of total mass of CH$_4$ entrapped in glacier-forefield sediments

### 2.3.1 Estimation for the GRF sampling zone

The mass of CH$_4$ ($m_{CH4}$) entrapped in a specific volume of porous sediment may be calculated using:

$$m_{CH4} = C_{CH4}\, \rho_{sed} \left[ A_{sed}\, T_{sed} \left( 1 - \theta_{t,sed} \right) \right] \tag{1}$$

where $C_{CH4}$ is sediment-entrapped CH$_4$ content (mass of CH$_4$ per mass of sediment), $\rho_{sed}$ is sediment-particle density, $A_{sed}$ and $T_{sed}$ are sediment-covered area and sediment thickness in the glacier forefield, and $\theta_{t,sed}$ is total sediment porosity. To determine $m_{CH4}$ for the GRF sampling zone, we applied Eq. (1) separately to each landform, but also used averaged values for entrapped CH$_4$ contents (from laboratory analyses), sediment thickness (from ERT field measurements), and sediment-covered area estimated from aerial maps (https://map.geo.admin.ch). In Eq. (1), the term in brackets represents the sediment's solid volume.





To compute the latter, we assumed a mean $\overline{\theta}_{t,sed}$ = 0.42 ± 0.02, as determined for this site by Nauer et al. (2012). To convert solid volume to sediment mass, a mean value of $\overline{\rho}_{sed}$ = 2.71 ± 0.15 g cm⁻³ was used, as derived by Daly (1935) from measurements of a variety of calcite rock samples.

    The total uncertainty in the estimated mean $CH_4$ mass $\overline{m}_{CH4}$, expressed as standard error (SE) of the mean ($\sigma_{\overline{m}_{CH4}}$), was computed using:

$$\sigma_{\overline{m}_{CH4}} = \overline{m}_{CH4} \sqrt{ \left( \frac{\sigma_{\overline{C}_{CH4}}}{\overline{C}_{CH4}} \right)^2 + \left( \frac{\sigma_{\overline{\rho}_{sed}}}{\overline{\rho}_{sed}} \right)^2 + \left( \frac{\sigma_{\overline{A}_{sed}}}{\overline{A}_{sed}} \right)^2 + \left( \frac{\sigma_{\overline{T}_{sed}}}{\overline{T}_{sed}} \right)^2 + \left( \frac{\sigma_{\overline{\theta}_{t,sed}}}{\overline{\theta}_{t,sed}} \right)^2 } \qquad (2)$$

where $\sigma_{\overline{x}}$ represents the SE associated with any parameter's mean value $\overline{x}$. The individual contribution of any parameter $x$ ($frac_x$, in %) to the total uncertainty in $\overline{m}_{CH4}$ was then computed using:

$$frac_x = \left( \frac{\overline{m}_{CH4}}{\sigma_{\overline{m}_{CH4}}} \left( \frac{\sigma_{\overline{x}}}{\overline{x}} \right) \right)^2 \times 100 \qquad (3)$$

    We note that throughout this manuscript SE values (reported as $\overline{x} \pm \sigma_{\overline{x}}$) are used as a measure of uncertainty of any

parameter's mean value $\overline{x}$, whereas standard deviations (SD, reported as $\overline{x}(\pm \sigma_x)$) are used as a measure of general parameter variability.

### 2.3.2 Estimation for the five glacier forefields (IMG, GRF, GRI, WIL, TSA)

    To compute total mass and associated uncertainty of sediment-entrapped $CH_4$ for all five glacier forefields, we employed Eqs. (1) and (2), but with partially modified parameters. For $C_{CH4}$ we used mean values of sediment-entrapped $CH_4$ contents

determined for each glacier forefield. In addition we determined mean values $\overline{A}_{sed}$ from estimates of the maximum and minimum extents of sediment-covered area within each glacier forefield. As maximum we used the areas exposed as a result of glacier retreat since the last glacial maximum (Little Ice Age, ~1850). The latter was estimated from the difference in glacial extent as taken from the most current (2018) and historic (~1850) topographic maps (Swisstopo; https://map.geo.admin.ch; Fig. S2). Minimum areas were directly estimated from the 2018 aerial maps. Also, data on sediment thickness was unavailable

for the IMG, GRI, WIL, and TSA glacier forefields, as well as for the GRF forefield outside of the designated sampling zone. We therefore used the average value of $T_{sed}$ = 10.0 ± 3.0 m obtained from our ERT measurements in the GRF sampling zone (see below) as an average $T_{sed}$ for all five glacier forefields. We note that our average $T_{sed}$ value agrees well with previous measurements performed in another Swiss glacier forefield, in which $T_{sed}$ ~8 m was obtained by borehole drilling (Kneisel and





Kääb, 2007). Finally, we used values of $\overline{\theta}_{t,sed}$ for GRF, GRI, and WIL forefields as determined for these sites by Nauer et al.

(2012). As such values were unavailable for the IMG and TSA forefields, we used a value of $\overline{\theta}_{t,sed}$ = 0.44 ± 0.05 for the latter, averaged from data reported for five calcareous glacier forefields (Nauer et al., 2012).

### 2.3.3 Estimation for sediments in all Swiss glacier forefields derived from calcareous bedrock

We again used Eq. (1) and (2) to upscale results and to compute a first estimate of the total mass of sediment-entrapped $CH_4$ contained in all Swiss glacier forefields derived from calcareous bedrock. In this case, we used the mean $\overline{C}_{CH4}$ of the five glacier

forefields. Calcareous glacier-forefield surface area in Switzerland ($A_{sed}$ in Eq. (1)) was estimated from available data on the decrease in glaciated area in the Swiss Alps between the Little Ice Age (~1850; Zemp et al. (2008)) and the year 2010 (Fischer et al., 2014), together with an estimate of the fraction of calcareous bedrock area to the total area of the Swiss Alps taken from the Tectonic Map of Switzerland 1:500.000 (Federal Office of Topography, swisstopo). Mean values for $\rho_{sed}$, $T_{sed}$, and $\theta_{t,sed}$ were used as described above.

## 3 Results

### 3.1 Geochemistry of gas entrapped in sediment and bedrock samples

Of the 271 sediment samples from the five glacier forefields we analyzed 256 samples for entrapped $CH_4$ contents. All analyzed sediments contained detectable amounts of $CH_4$ ranging from 0.08 to 73.81 µg $CH_4$ g$^{-1}$ dry weight (d.w.; Fig. 3), with an average of 14.9 (± 17.0) µg $CH_4$ g$^{-1}$ d.w.. Gas released from 225 samples was analyzed for $C_2H_6$ and $C_3H_8$ contents, of which

215 contained detectable amounts of $C_2H_6$ ranging from 0.002 to 1.67 µg $C_2H_6$ g$^{-1}$ d.w., with an average of 0.25 (± 0.32) µg $C_2H_6$ g$^{-1}$ d.w.. In addition, 146 out of 225 samples contained detectable amounts of $C_3H_8$ ranging from 0.001 to 0.82 µg $C_3H_8$ g$^{-1}$ d.w., with an average of 0.11 (± 0.15) µg $C_3H_8$ g$^{-1}$ d.w. (not shown).

The average gas-wetness ratio for all sediment samples was 75.2 (± 48.4), and the average $\delta^{13}C_{CH4}$ was -28.23 (± 3.42) ‰. Plotting $\delta^{13}C_{CH4}$ values vs. gas-wetness ratios in a so-called Bernard diagram (Fig. 4; Bernard et al. (1978)) indicated a

thermogenic origin for sediment-entrapped $CH_4$, derived from ancient terrestrial or marine organic matter (kerogen types III and II, Fig. 4). Although $CH_4$ extracted from sediments collected in the IMG glacier forefield showed a higher variability in gas-wetness ratios than $CH_4$ extracted from sediments of other glacier forefields, it still fell into the same origin type in the Bernard diagram.

All 72 bedrock samples were analyzed for $CH_4$ content, and 64 contained detectable amounts of $CH_4$ ranging from 0.06

to 108.58 µg $CH_4$ g$^{-1}$, with an average of 11.4 (± 20.0) µg $CH_4$ g$^{-1}$ (Fig. 3). The average $\delta^{13}C_{CH4}$ value of -29.21 (± 2.77) ‰ was similar to that of sediment-entrapped $CH_4$. Likewise, the average gas-wetness ratio of gas extracted from rocks was 78.45 (± 121.84), similar in value but with higher variability than gas-wetness ratios for sediment-entrapped $CH_4$ (Fig. 4). Together,



these data suggest a common, thermogenic origin of entrapped $CH_4$ in sediments and rocks, with little apparent alteration from physical/chemical weathering. Moreover, our data suggest that entrapped $CH_4$ is of similar origin in all five glacier forefields.

### 3.2 Spatial distribution of sediment-entrapped CH₄ contents in the GRF sampling zone

Methane contents in 52 samples collected from 20–70 cm depth ranged from 1.19 to 11.24 µg $CH_4$ $g^{-1}$ d.w., with one exceptionally high value at 40 cm depth (Fig. 5). Based on these data, there was no clear correlation between sediment depth and entrapped $CH_4$ contents (one-way ANOVA, $p = 0.9$). Thus, we subsequently proceeded to collect sediments from 20 cm depth only, and assumed these samples to be representative in terms of entrapped $CH_4$ content for the entire sediment thickness.

The effects of sediment age and landform on entrapped $CH_4$ contents were tested using sediments collected from 20 cm depth at 99 locations (Fig. 2a). The $CH_4$ contents in these samples ranged from 0.59 to 34.82 µg $CH_4$ $g^{-1}$ d.w. (Fig. 3b), with an average of 5.30 (± 4.86) µg $CH_4$ $g^{-1}$ d.w.. Two-way ANOVA analysis indicated that landform had a significant effect on sediment-entrapped $CH_4$ contents ($p = 0.03$), whereas effects of sediment age ($p = 0.19$) and the combined effects of sediment age and landform on entrapped $CH_4$ contents ($p = 0.37$) were insignificant. Post-hoc comparisons using the Tukey HSD test indicated that mean values for sediment-entrapped $CH_4$ content (Table 1) were significantly different between floodplain and sandhill ($p = 0.03$), and weakly different between floodplain and terrace ($p = 0.10$). The difference between terrace and sandhill with respect to mean sediment-entrapped $CH_4$ content was insignificant ($p = 0.88$).

### 3.3 Mass of sediment-entrapped CH₄ in the GRF sampling zone

To estimate the mass of sediment-entrapped $CH_4$ stored within the GRF sampling zone, we used Eq. (1) with mean values on entrapped $CH_4$ contents, sediment thickness, and sediment-covered area determined for each of the three landforms (Table 1). Whereas $\overline{C}_{CH4}$ varied by a factor <1.4 between landforms, sediment thickness was highly variable along the five measured ERT profiles (range 1.0–31.5 m; Fig. 6, Fig. S3), and $\overline{T}_{sed}$ varied by a factor of ~2 between landforms (Table 1). Sediment-covered area also showed substantial variation between the different landforms. Within the GRF sampling zone, the sandhill landform comprised the largest sediment-covered area with $\overline{A}_{sed} \approx 10^5$ $m^2$, about 5 times larger than for floodplain and terrace. Consequently, the largest sediment mass was contained in the sandhill landform (factor 2–3 larger than floodplain and terrace, Table 1). All three landforms combined featured a surface area of ~$1.5 \times 10^5$ $m^2$, and contained an estimated mass of ~$2.3 \times 10^6$ t sediment. Adding up the masses of sediment-entrapped $CH_4$ for each landform yielded a total $\overline{m}_{CH4} = 9.7 \pm 3.0$ t $CH_4$. When calculated using average values for entrapped $CH_4$ contents, sediment thickness, and sediment-covered area, the estimated $\overline{m}_{CH4}$ within the GRF sampling zone was $12.3 \pm 3.9$ t $CH_4$ (last row in Table 1). Uncertainties in individual $\overline{m}_{CH4}$ up to ~50% mostly arose from uncertainties in $\overline{T}_{sed}$ and, to a smaller degree, $\overline{C}_{CH4}$.



### 3.4 Contents and total mass of sediment-entrapped $CH_4$ in five glacier forefields

Methane contents varied substantially between different glacier forefields (Table 2), with distance between the forefields playing an apparently important role. Specifically, the IMG, GRF, and GRI glacier forefields are located in the Northeast of the Helvetic Nappes relatively close to each other (Fig. 1) and featured similar, low sediment-entrapped $CH_4$ contents.

Likewise, the WIL and TSA glacier forefields are located close to each other in the Southwest of the Helvetic Nappes and featured similar, but high sediment-entrapped $CH_4$ contents. Indeed, our ANOVA results indicated that differences in sediment-entrapped $CH_4$ contents were insignificant between the IMG, GRF, and GRI glacier forefields ($p = 0.36$) and between the WIL and TSA glacier forefields ($p = 0.18$). Conversely, differences in entrapped $CH_4$ contents between the two groups of glacier forefields were highly significant ($p < 0.0001$).

The total mass of $CH_4$ entrapped in sediments of the five glacier forefields was calculated using estimated values for sediment thickness ($10.0 \pm 3.0$ m; the thickness measured in the GRF sampling zone (see above)) and sediment-particle density ($2.71 \pm 0.15$ g/cm³; Daly (1935)) that were assumed identical for all five forefields, as well as specific data for each glacier forefield on entrapped $CH_4$ contents, sediment-covered area, and sediment porosity (Table 2). Whereas $\overline{\theta}_{t,sed}$ values varied only little between the five forefields, $\overline{A}_{sed}$ varied up to a factor of ~3 (IMG vs. WIL), and $\overline{C}_{CH4}$ up to a factor of ~7 (GRF vs.

WIL). This led to substantial variability in the estimated total mass of sediment-entrapped $CH_4$ between the five forefields, which ranged from $200 \pm 74$ t $CH_4$ for the GRF glacier forefield to $3881 \pm 1367$ t $CH_4$ for the WIL forefield (Fig. 7a). Estimates of sediment-entrapped $CH_4$ for the WIL and TSA glacier forefields were significantly larger than for IMG, GRF, and GRI. For all five forefields, sediment thickness and sediment-covered area contributed most to uncertainties in the quantification (Fig. 7b). Entrapped $CH_4$ contents, sediment porosity, and sediment-particle density contributed little to the calculated uncertainties.

**3.5 Mass of sediment-entrapped $CH_4$ in all Swiss glacier forefields on calcareous bedrock**

The first estimate of the total mass of sediment-entrapped $CH_4$ in all calcareous Swiss glacier forefields was based on published data on glacier retreat in the Swiss Alps, an estimation of the fraction of calcareous glacier-forefield surface area, mean values for sediment thickness, sediment-particle density and total sediment porosity, as well as a mean value for sediment-entrapped $CH_4$ content obtained from the five investigated glacier forefields ($18.5 \pm 4.4$ µg $CH_4$ g⁻¹ d.w.; Table 3). Between the end of

the Little Ice Age (~1850) and 2010, the glaciated area within the Swiss Alps has decreased by ~676 km² to less than 60 % of its original value (data sources see Table 3). When multiplied by the fraction of calcareous bedrock area in the Swiss Alps ($54.6 \pm 1.7$ %), this yielded an exposed calcareous glacier-forefield area of ~369 km². The total sediment mass contained within this exposed calcareous glacier-forefield area was then computed as $5.62 \times 10^9 \pm 1.46 \times 10^9$ t. From these numbers, the total mass of sediment-entrapped $CH_4$ in all Swiss glacier forefields derived from calcareous bedrock was computed as $1.04 \times 10^5 \pm$

$3.7 \times 10^4$ t $CH_4$.



## 4 Discussion

### 4.1 Widespread occurrence of sediment-entrapped, thermogenic CH₄ in calcareous glacier forefields

We detected substantial quantities of sediment-entrapped $CH_4$ in all sampled glacier forefields. Entrapped $CH_4$ was ubiquitously encountered at different sediment depths, and in different forefield landforms and sediment-age classes. We also

detected entrapped $CH_4$ in most bedrock samples obtained from these glacial catchments. Furthermore, our data indicated that both sediment- and rock-entrapped $CH_4$ are of thermogenic origin. Thus, the results presented here extend our previous studies (Nauer et al., 2012; Zhu et al., 2018) by providing a more detailed survey on entrapped $CH_4$ contained in glacier-forefield sediments across the Helvetic Nappes, and support our hypothesis on its widespread occurrence and thermogenic origin in calcareous, Swiss Alpine glacier forefields. On the other hand, we cannot entirely reject the possibility for the presence of

microbial $CH_4$ sources in certain parts of glacier forefields, particularly in water-logged sediments. Methanogenic potential in isolated hotspots of water-logged sediments was previously confirmed for the WIL glacier forefield, but considered to be of minor importance under field conditions (Zhu et al., 2018). In the present study, no attempt was made to specifically identify potential methanogenic hotspots in sediments of the other four glacier forefields.

Methane is commonly found in organic-rich sedimentary rocks such as shales, marls, and limestones as a product of

the thermal maturation of buried organic matter (Etiope, 2017; Horsfield and Rullkötter, 1994). Previous studies on fluid inclusions in quartz and calcite minerals collected from Alpine fissures and veins within the Helvetic Nappes revealed the existence of four fluid zones, including a large thermogenic $CH_4$ zone (Gautschi et al., 1990; Mazurek et al., 1998; Mullis et al., 1994; Tarantola et al., 2007). The five glacier forefields we sampled in this study were all located within or near the border of this thermogenic $CH_4$ zone (see Fig. 1 in Tarantola et al. (2007)). Our results thus agree with previous findings on the

occurrence of thermogenic $CH_4$ in this region, including the occurrence of thermogenic $CH_4$ detected in gas seeps near Giswil, Central Switzerland, which lies on Penninic Flysch underlain by Helvetic Nappes (Etiope et al., 2010). On the other hand, our results also show that $CH_4$ entrapment within the Helvetic Nappes is not restricted to fluid inclusions in fissure minerals, but that substantial quantities of $CH_4$ are entrapped within the matrix of the sedimentary bedrock and sediment particles themselves, presumably within inter- and intragranular macro- and microporosity (Hashim and Kaczmarek, 2019; Moshier,

1989; Léonide et al., 2014; Abrams, 2017).

Our geochemical data further indicate a common origin for $CH_4$ entrapped in bedrock and glacier-forefield sediments, derived from ancient terrestrial and marine organic matter (kerogen types III and II, respectively; Fig. 4). This provides further evidence that $CH_4$ entrapped in the forefield sediments of the Helvetic Nappes has its origin in the calcareous parent bedrock. Moreover, terrestrial and marine organic matter as the ultimate source of sediment- and rock-entrapped $CH_4$ agrees with the

origin of the Helvetic Nappes: their sediments and organic matter were originally deposited under highly variable climatic conditions on the shallow northern shelf of the ancient Alpine Tethys Ocean (Weissert and Mohr, 1996; Weissert and Stössel, 2015).



### 4.2 Spatial distribution of sediment-entrapped CH$_4$ within and between glacier forefields

Sediment-entrapped CH$_4$ contents showed moderate variability within each glacier forefield (Fig. 3a-e). As sediments were
largely derived by glacial erosion from the surrounding calcareous bedrock (Chesworth et al., 2008; Fu and Harbor, 2011), the
observed variability in sediment-entrapped CH$_4$ contents reflects the variability in entrapped CH$_4$ contents of the various
geological formations present in each catchment (Fig. 3f). Entrapped CH$_4$ contents in sedimentary bedrocks is typically
affected by three main factors: the quantity and quality of organic matter buried during sediment deposition, the thermal history
during sediment diagenesis and subsequent organic matter catagenesis, and the resulting permeability of the calcareous
bedrock, which affects potential gas migration (e.g., Dayal, 2017; Horsfield and Rullkötter, 1994; Mani et al., 2017). Whereas
geological formations contained within the same nappe are expected to possess a similar thermal history, the quantity and
quality of organic matter buried may vary substantially between individual formations depending on prevailing conditions
during the period of sediment deposition (Weissert and Mohr, 1996; Weissert et al., 1985). Thus, variability in rock- and
sediment-entrapped CH$_4$ contents is to be expected for glacial catchments featuring geological formations from different time
periods, as was observed for all of the glacier forefields sampled in this study (Table 2).

Our study in the GRF forefield sampling zone indicated that sediment-entrapped CH$_4$ content varied little with sediment
depth (Fig. 5) and sediment age. However, we cannot exclude the possibility that such variations could be somewhat larger
outside of the sampled depth interval, e.g., in top-layer sediments at depths < 5 cm as a result of enhanced chemical, physical,
or biological weathering (Bernasconi et al., 2011; van der Meij et al., 2016; Lazzaro et al., 2009). We refrained from collecting
top-layer sediments because in all five glacier forefields they were generally much coarser and thus did not appear
representative of bulk sediments present at greater depth. We assume that sediment fines are continuously removed from the
top layer as a result of physical (wind and water) erosion.

On the other hand, we consider the lack of significant variation with sediment age as an indication that CH$_4$ in glacier-
forefield sediments is relatively stable in its entrapped state. This hypothesis is supported by results of our geochemical
analyses for all five glacier forefields, which mostly indicated high similarity between sediment- and rock-entrapped CH$_4$ in
terms of the range of measured CH$_4$ contents (Fig. 3f), as well as gas-wetness ratios and $\delta^{13}C_{CH4}$ values (Fig. 4). Thus, although
sediments have likely undergone great alteration during and after erosion from the parent bedrock, changes in entrapped CH$_4$
geochemical characteristics appeared negligible. This indicates that a potential release of entrapped CH$_4$ from sediment
particles by molecular diffusion, or oxidation of CH$_4$ in its entrapped state within sediment particles, should be of minor
importance, as these processes would be expected to cause a noticeable change in CH$_4$ geochemical characteristics (Schloemer
and Krooss, 2004; Whiticar, 1999; Zhang and Krooss, 2001). Our findings therefore suggest that CH$_4$ entrapped in bedrock
and sediment matrices resides largely in inaccessible, occluded rather than connected pore spaces. However, a potential release
of entrapped CH$_4$ from occluded pore spaces may yet occur via sediment erosion processes, in particular by means of physical
and/or chemical weathering of calcareous minerals (Emmanuel and Levenson, 2014; Ryb et al., 2014; Trudgill and Viles,
1998). As these processes act on rock surfaces, they are of great important to sediments with large specific surface areas, the



latter being inversely related to particle size (Michel and Courard, 2014). Although we are aware that similar erosion processes will act upon large bedrock surfaces, e.g., rock walls and other outcrops within glacial catchments, we have so far refrained from considering $CH_4$ release from these locations because of the much smaller specific surface areas involved. Unfortunately, the release of entrapped $CH_4$ as a result of sediment erosion may not be detectable in the sediment's entrapped $CH_4$ contents,
as both $CH_4$ and sediment mass is lost as a result of erosion. Hence, our present data set yields no information on the relevance of erosion processes for $CH_4$ release.

In contrast to sediment depth and sediment age, we detected a small but significant difference in mean sediment-entrapped $CH_4$ content between landforms within the GRF sampling zone. Specifically, mean entrapped $CH_4$ content in floodplain sediments was significantly higher than in terrace and sandhill sediments (Table 1). We can only speculate about
possible reasons for this observation. One reason could be that floodplain sediments, intermittently removed and deposited by the glacial stream during and after flooding events, originate from locations far outside of our sampling zone, where sampling of the parent bedrock, e.g., from steep rock walls, was not feasible (Fig. S2). It is therefore possible that we missed to sample parent bedrock types with high entrapped $CH_4$ contents in this or any of the other glacial catchments.

Finally, our data revealed large regional differences in mean sediment-entrapped $CH_4$ contents between glacier
forefields (Table 2). This may be explained by the fact that sediments in glacier forefields located in close proximity to one another are, at least in part, derived from the same individual nappes and geological formations contained therein. For example, both the WIL and TSA glacier forefields harbor sediments derived from the Wildhorn nappe, featuring several identical geological formations. Hence, this result supports our previous hypothesis that differences in lithology and tectonic settings between individual nappes play an important role in determining bedrock- and thus sediment-entrapped $CH_4$ contents (Zhu et
al., 2018). Regional differences in entrapped $CH_4$ contents paired with differences in sediment-covered area led to significant variation in the estimates for total mass of $CH_4$ stored in sediments of the five glacier forefields (Fig. 7a). Uncertainties associated with these estimates were reasonably small, and arose largely from uncertainties in sediment thickness and sediment-covered area (Fig. 7b). To further reduce these uncertainties, measurements of these parameters across entire glacier forefields would be of help using, e.g., geophysical methods for sediment thickness (such as the ERT method used in the GRF
sampling zone), and field mapping of sediment-covered area in combination with GIS based methods utilizing digital elevation models (e.g., Geilhausen et al., 2012; Smith and Clark, 2005; Zemp et al., 2005). Unfortunately, field measurements in the rugged alpine environment are typically time-consuming, expensive, and challenging to perform.

**4.3 A substantial quantity of sediment-entrapped $CH_4$ with yet unknown fate**

Our first, rough estimate for the total quantity of $CH_4$ entrapped in sediments of all calcareous Swiss glacier forefields
combined yielded a substantial mass of $1.04×10^5 ± 3.7×10^4$ t $CH_4$, contained within a solid volume of ~2.1 km$^3$ glacier-forefield sediments. At first glance, this number appears large when compared with estimates of annual $CH_4$ release from lake sediments into the lower, anoxic water column of a Swiss lake ($1.7×10^3$ t $CH_4$; Schubert et al. (2010)), and annual $CH_4$ emissions to the atmosphere ($5.7×10^3$ t $CH_4$) from all natural and semi-natural sources in Switzerland, including emissions from lakes,



reservoirs, wetlands, and wild animals (Hiller et al., 2014). However, whereas the latter data represent annual $CH_4$ fluxes, the
fate of sediment-entrapped $CH_4$ remains elusive to date (see below). On the other hand, our number is in good agreement with
a previous estimate on $CH_4$ content for Valanginian marl, a geological formation within the Helvetic Nappes, containing calcite
fracture fill ($\sim0.7\times10^5$–$2.1\times10^5$ t $CH_4$ km$^{-3}$ bedrock; Gautschi et al. (1990)).

Our estimate for total sediment-entrapped $CH_4$ mass is subject to substantial uncertainty. The two largest contributors
to the calculated uncertainty are sediment-entrapped $CH_4$ content and sediment depth. In addition, there is considerable
uncertainty in the exposed calcareous glacier-forefield area, as the latter was only roughly estimated based on glacier retreat
and the fraction of calcareous bedrock area in the Swiss Alps. As discussed above for individual glacier forefields, field
measurements and GIS based methods may help to reduce uncertainties related to sediment depth and exposed area. An
important way to reduce uncertainty related to entrapped $CH_4$ contents would be to generate a database of $CH_4$ contents for
different geological formations present within the Helvetic Nappes, as lithology and tectonic settings appear to control $CH_4$
contents. Determination of the areal extent of different geological formations would likely help to reduce uncertainties in
sediment-entrapped $CH_4$ mass.

Whether or not sediment-entrapped $CH_4$ plays a role as an emission source to the atmosphere will largely depend upon
its rate of release from sediment particles and its potential consumption by MOBs in aerated sediments. Whereas we produced
some evidence that $CH_4$ is stable in its entrapped state (see discussion above), further investigations will be required to
specifically elucidate mechanisms and fluxes of $CH_4$ release in forefield sediments, in particular during periods of enhanced
physical/chemical weathering, e.g., during rainstorms or snow melt (Winnick et al., 2017). On the other hand, atmospheric
$CH_4$ oxidation was previously detected in several glacier forefields including our GRF site (Bárcena et al., 2011; Chiri et al.,
2015; Hofmann et al., 2013). These studies indicated that MOB activity in forefield sediments establishes quickly (within the
first 10 years after glacier retreat), and fluxes of $CH_4$ uptake from the atmosphere increase to values comparable to mature
soils within a few decades (Chiri et al., 2015). Nonetheless, intermittent $CH_4$ emissions to the atmosphere were also observed
in GRF floodplain sediments (Chiri et al., 2017). Hence, we hypothesize that $CH_4$ released from sediment particles may be
consumed by MOB, at least under favorable environmental conditions, and serve as an additional source of energy and carbon
to this group of microorganisms. This hypothesis, of course, awaits experimental confirmation.

**5 Summary and Conclusions**

Our results provide new evidence for the widespread occurrence of sediment-entrapped, thermogenic $CH_4$ in Swiss calcareous
glacier forefields. As entrapped $CH_4$ with highly similar geochemical characteristics was also detected in most bedrock samples
collected from nearby geological formations, we conclude that $CH_4$ entrapped in forefield sediments of the Helvetic Nappes
has its origin in the calcareous parent bedrock. Hence, spatial variability in sediment-entrapped $CH_4$ contents within glacier
forefields largely reflects the variability in entrapped $CH_4$ contents of the surrounding bedrock types.



Within glacier forefields, sediment-entrapped CH₄ contents and other geochemical characteristics showed little systematic variation with sediment age and thus time of exposure to the atmosphere following glacier retreat. Together with the noted similarity in geochemical characteristics we took this finding as evidence that CH₄ in glacier-forefield sediments is relatively stable in its entrapped state, presumably because it resides in occluded pore spaces within bedrock and sediment matrices. This further indicates that CH₄ entrapment within the Helvetic Nappes is not restricted to fluid inclusions in fissure

minerals, but that substantial quantities of CH₄ are entrapped within the matrix of the sedimentary bedrock and sediment particles themselves. On the other hand, our results revealed large regional differences in mean sediment-entrapped CH₄ contents between glacier forefields, supporting our previous hypothesis that differences in lithology and tectonic settings between individual nappes play an important role in determining bedrock- and thus sediment-entrapped CH₄ contents.

       Our first estimate for the total quantity of CH₄ entrapped in sediments of all calcareous Swiss glacier forefields suggests

the presence of a substantial CH₄ mass. Whereas we have provided evidence for its stability in its entrapped state, we cannot exclude the possibility that sediment-entrapped CH₄ is being emitted into the sediments' pore space as a result of physical or chemical weathering. Whether this would lead to emissions into the atmosphere will largely depend upon the rate of release from sediment particles and its potential consumption by MOBs in aerated sediments. Experiments are needed and currently ongoing in our laboratory to quantify these two processes under variable environmental conditions.


*Data availability*. The data used in this manuscript will be made available on ETH Zurich Research Collection after the manuscript is published.

*Supplement*. The supplement related to this article is available online at: …..


*Author contribution*. BZ, MR, MK, and MHS helped with sample collection and/or geochemical measurements, and substantially contributed to the interpretation of data. DB helped with ERT measurements and subsequent ERT data analyses. BZ and MHS wrote the manuscript. MHS designed the study and acquired the funding for the project. All authors commented on the manuscript and approved the final version of the manuscript.


*Competing interests*. The authors declare that they have no conflict of interest.

*Acknowledgements*. We are grateful to S. Bernasconi (ETHZ) and C. Schubert (EAWAG) for assistance with stable isotope analyses, and to S. Bernasconi for providing valuable suggestions to an early version of this manuscript. We thank L. Baron

(UNIL) for introducing us to initial ERT measurements in glacial systems, and P. Erickson, M. Meola, and S. Meyer (all ETHZ) for their help during ERT field campaigns. We also thank C., F., and M. Stadler (Altdorf, UR) for granting road access to the GRF site.





*Financial support.* This research project was funded by the Swiss National Science Foundation under grant no.
200021_153571. Additional funding was received from ETH Zurich.

*Review statement.* This paper was edited by xyz and reviewed by xyz referees.

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

710





**Figures**

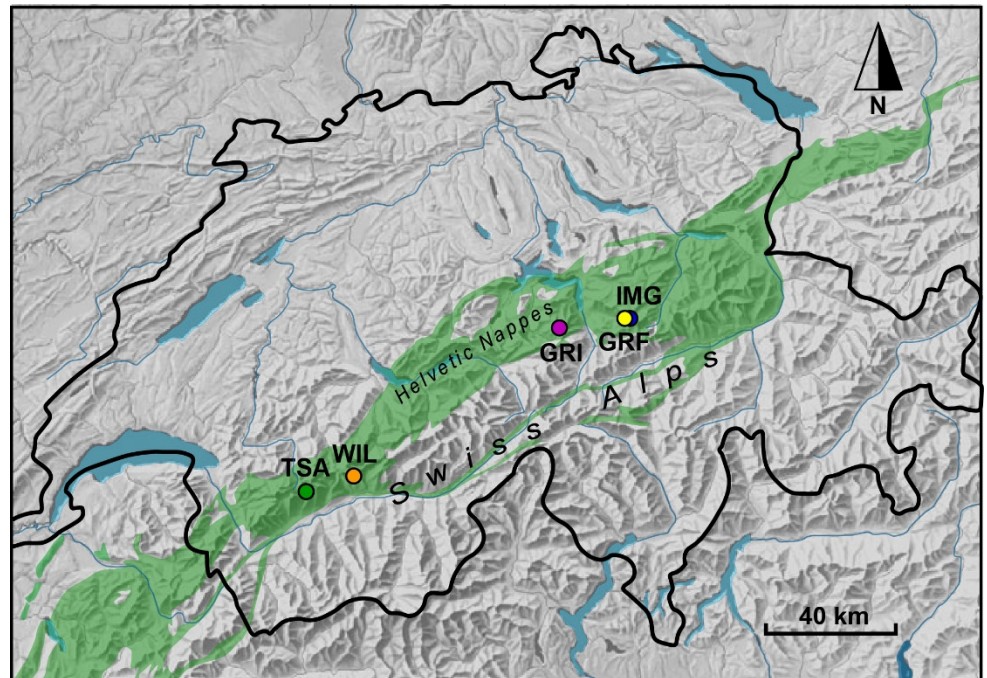

**Figure 1.** Map of Switzerland showing five glacier forefields from which sediment and bedrock samples were collected (Central Switzerland:
715  Im Griess, IMG; Griessfirn, GRF; Griessen, GRI; Canton Valais: Wildstrubel, WIL; Tsanfleuron, TSA). All forefields are located within
the Helvetic Nappes (green-shaded area), which consist largely of Mesozoic limestones, shales, and marls (map modified from Weissert and
Stössel (2015)).





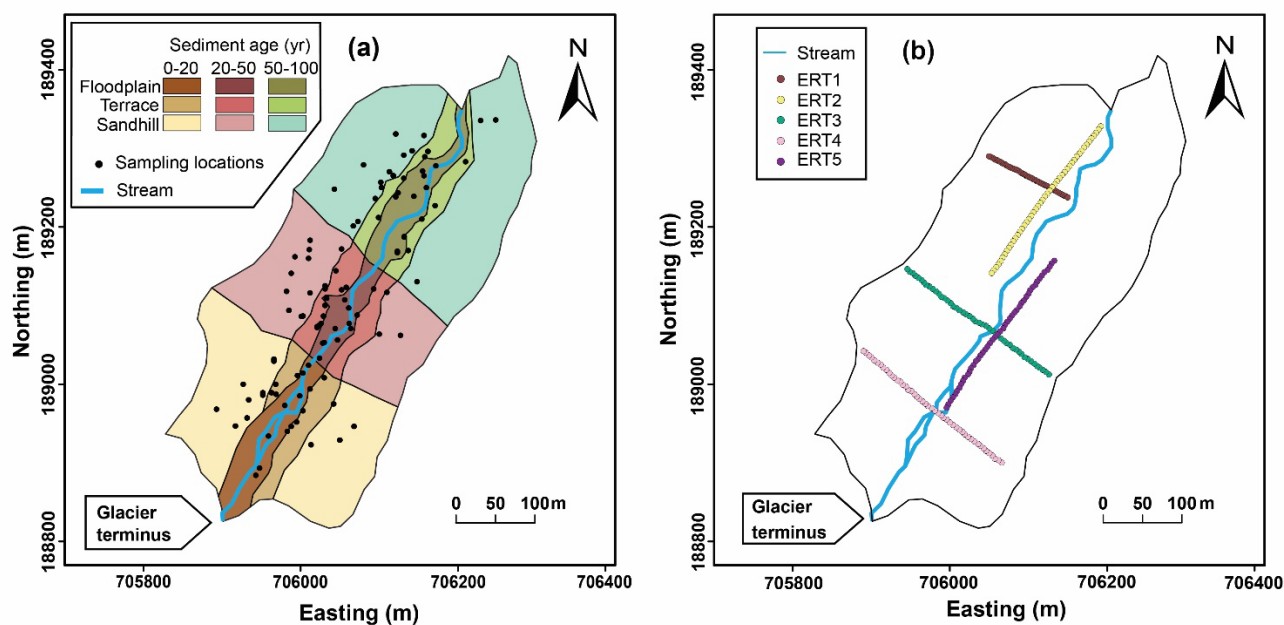

**Figure 2.** Sampling zone at Griessfirn (GRF) glacier forefield showing (a) blocks and sampling locations to study the effect of sediment age and glacier-forefield landforms on entrapped CH$_4$ contents, and (b) locations of five electrical resistivity tomography (ERT) profiles to measure sediment thickness. Axes show the Swiss CH1903/LV03 coordinate system (units in meters).



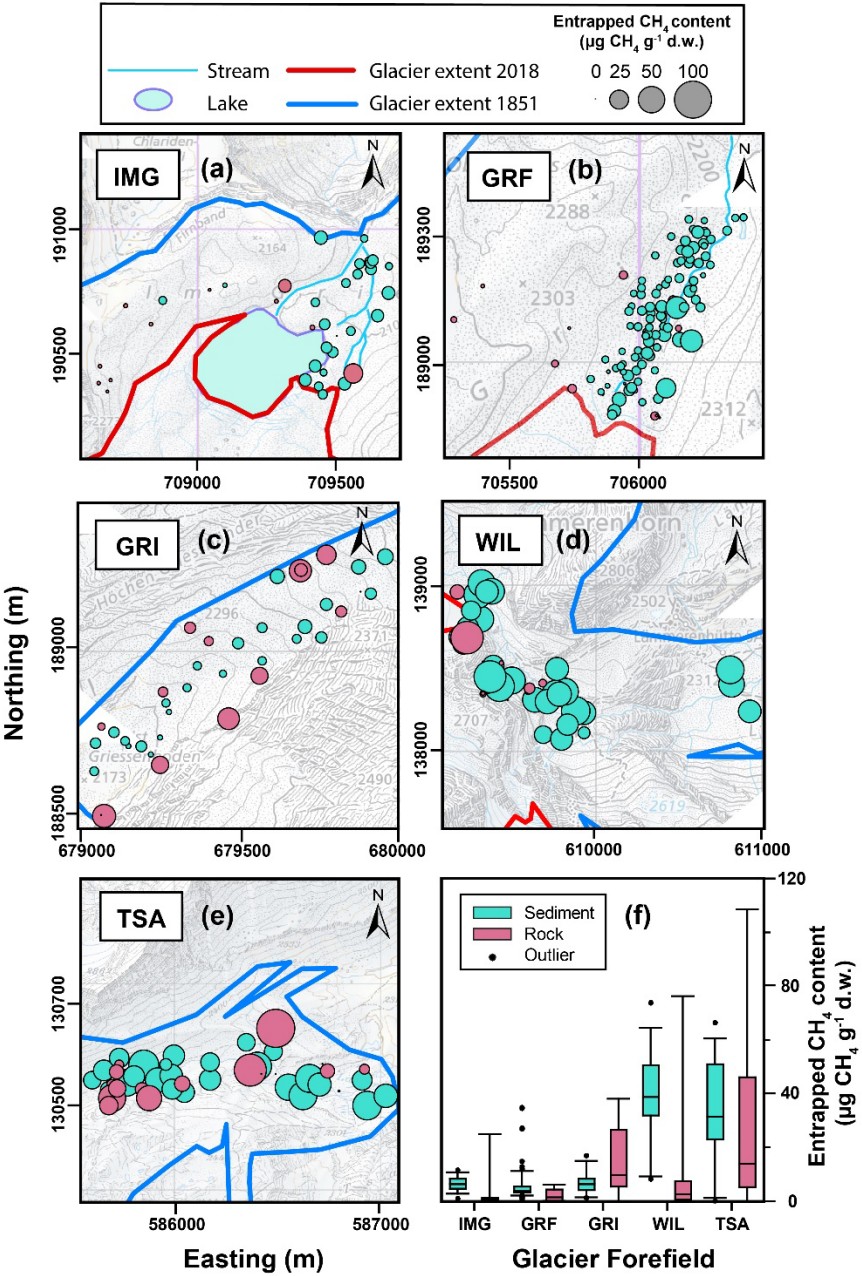

**Figure 3.** Spatial distribution of entrapped CH₄ contents in sediments (blue bubbles) and rocks (red bubbles) collected from (a) Im Griess (IMG), (b) Griessfirn (GRF), (c) Griessen (GRI), (d) Wildstrubel (WIL), and (e) Tsanfleuron (TSA) glacier forefields (bubble size proportional to entrapped CH₄ content). Background elevation data modified from swisstopo (Swiss Federal Office of Topography; maps.geo.admin.ch); axes show the Swiss CH1903/LV03 coordinate system (units in meters). (f) Box-whisker plot showing the range of entrapped CH₄ contents in sediments and rocks for each glacier forefield. Boxes represent 25th, 50th (median), and 75th percentile; whiskers indicate 5th and 95th percentile, outliers are marked as dots.



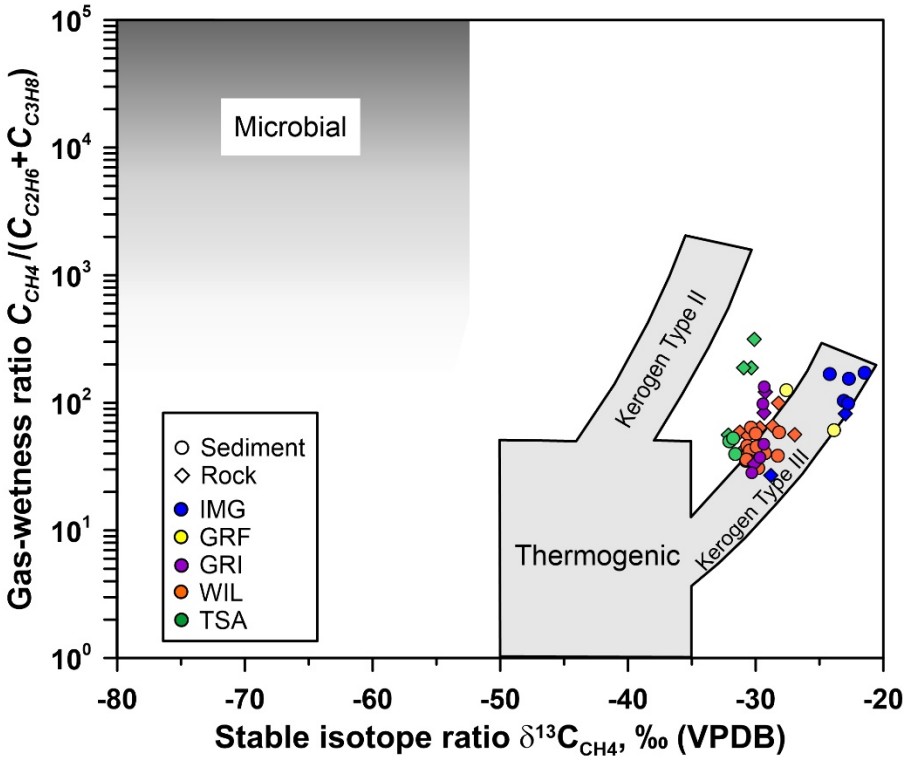

**Figure 4.** Adapted Bernard diagram (Bernard et al., 1978) showing gas-wetness ratio ($C_{CH4} / (C_{C2H6} + C_{C3H8})$)) versus $\delta^{13}C_{CH4}$ for gas released from selected sediment and rock samples collected from Im Griess (IMG), Griessfirn (GRF), Griessen (GRI), Wildstrubel (WIL), and Tsanfleuron (TSA) glacier forefields. Grey-shaded areas indicate different $CH_4$ origins (microbial vs. thermogenic).





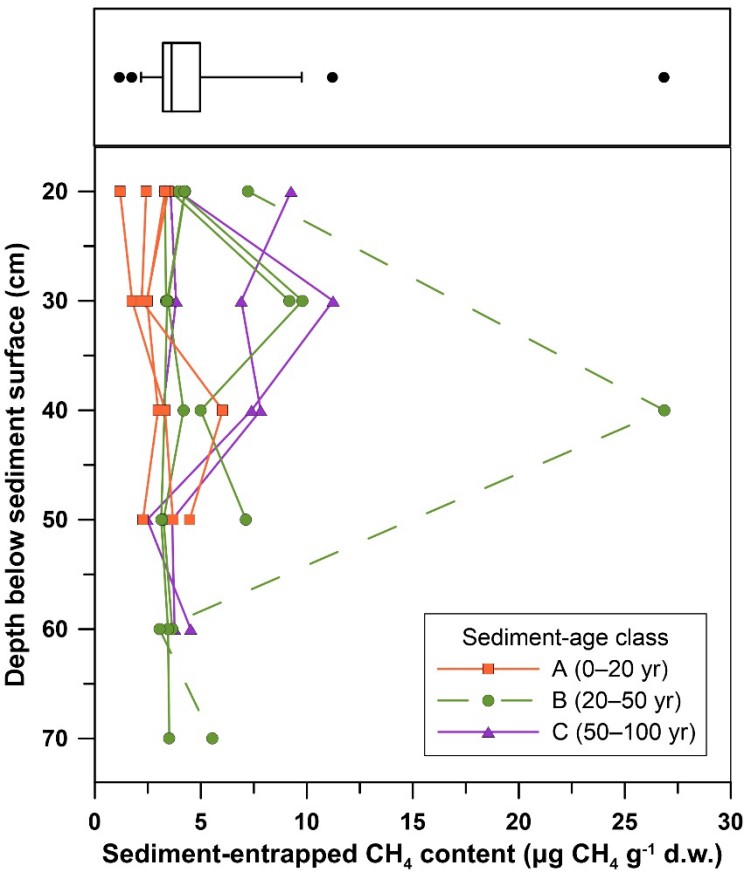

**Figure 5.** Sediment-entrapped $CH_4$ contents as a function of sediment depth for samples collected in three sediment-age classes in the Griessfirn (GRF) sampling zone. The box-whisker plot on top shows the range of entrapped $CH_4$ contents displayed below, with the box representing 25th, 50th (median), and 75th percentile; whiskers indicate the 5th and 95th percentile, outliers are marked as dots.





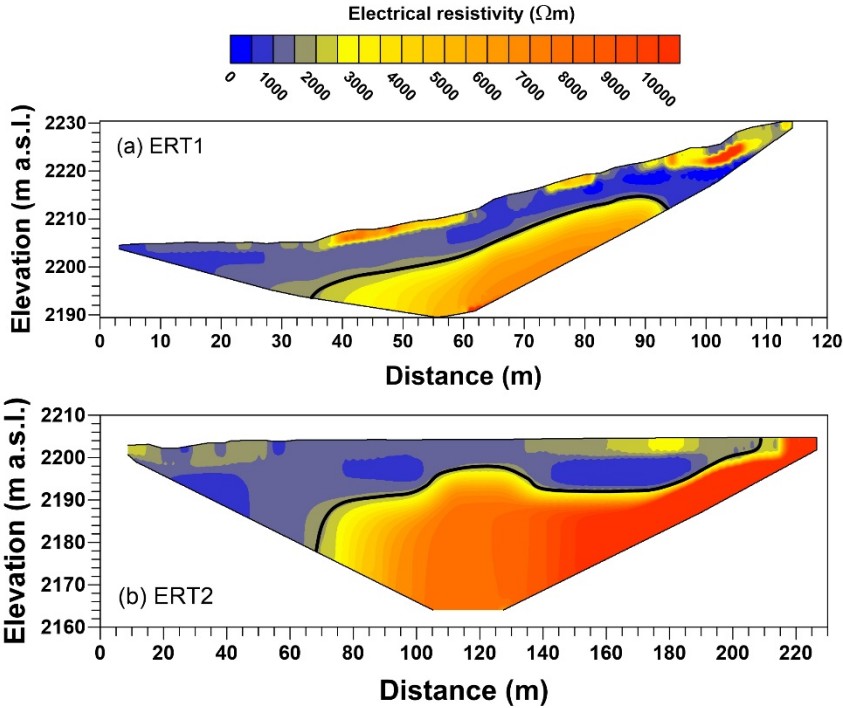

**Figure 6.** Vertical, two-dimensional electrical-resistivity-tomography (ERT) cross sections of profiles (a) ERT1 and (b) ERT2 collected in the sampling zone of Griessfirn (GRF) glacier forefield. Solid black lines indicate the approximate location of the interface between unconsolidated sediment and the bedrock underneath. Lines were omitted at locations where the sediment-rock interface was too deep to be detected.

750





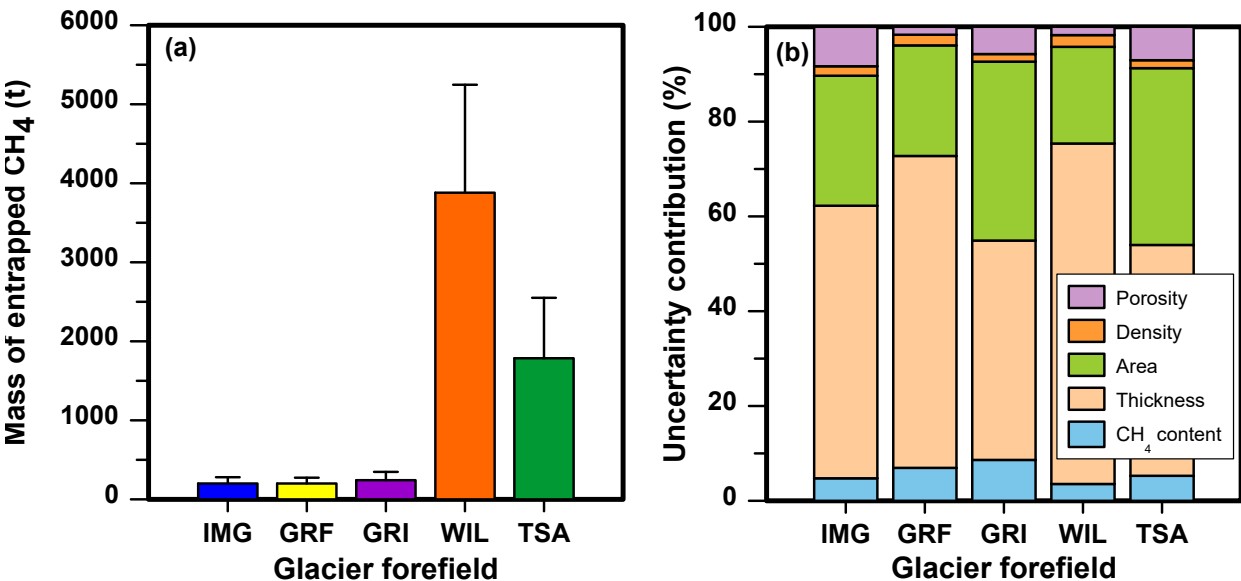

**Figure 7.** a) Estimated mass of CH$_4$ entrapped in sediments of Im Griess (IMG), Griessfirn (GRF), Griessen (GRI), Wildstrubel (WIL), and Tsanfleuron (TSA) glacier forefields. b) Contribution of sediment-entrapped CH$_4$ contents, sediment thickness and area, sediment-particle density, and total porosity to total uncertainties in the estimation of the mass of sediment-entrapped CH$_4$ (error bars in a)) for each of the five glacier forefields.

760





## Tables

**Table 1.** Mean values and uncertainties for sediment-entrapped $CH_4$ content, sediment thickness, sediment-covered area, sediment mass, and estimated mass of entrapped $CH_4$ in three different landforms of the Griessfirn (GRF) glacier-forefield sampling zone.

| Landform | Entrapped $CH_4$ content ($\mu g\ CH_4\ g^{-1}$ d.w.) | Sediment thickness (m) | Sediment-covered area ($m^2$) | Sediment mass (t sed.) | Entrapped $CH_4$ mass (t $CH_4$) |
|---|---|---|---|---|---|
| | $\overline{C}_{CH4} \pm \sigma_{\overline{C}_{CH4}}$ [a] | $\overline{T}_{sed} \pm \sigma_{\overline{T}_{sed}}$ | $\overline{A}_{sed} \pm \sigma_{\overline{A}_{sed}}$ | $\overline{m}_{sed} \pm \sigma_{\overline{m}_{sed}}$ | $\overline{m}_{CH4} \pm \sigma_{\overline{m}_{CH4}}$ |
| Floodplain | 6.37 ± 0.55 | 11.8 ± 3.0 | $2.07×10^4 \pm 2.0×10^2$ | $3.84×10^5 \pm 1.0×10^5$ | 2.4 ± 0.7 |
| Terrace | 4.72 ± 0.97 | 12.5 ± 4.0 | $2.06×10^4 \pm 2.0×10^2$ | $4.04×10^5 \pm 1.3×10^5$ | 1.9 ± 0.7 |
| Sandhill | 5.04 ± 0.78 | 6.4 ± 3.2 | $1.05×10^5 \pm 1.0×10^3$ | $1.06×10^6 \pm 5.4×10^5$ | 5.4 ± 2.8 |
| | | | | | 9.7 ± 3.0 [b] |
| Combined | 5.30 ± 0.49 | 10.0 ± 3.0 | $1.47×10^5 \pm 1.4×10^3$ | $2.31×10^6 \pm 7.9×10^5$ | 12.3 ± 3.9 [c] |

[a] standard error of the mean (SE).

[b] calculated by adding up estimated mass of entrapped $CH_4$ from each landform.

[c] calculated using average values for entrapped $CH_4$ contents, sediment thickness, and sediment-covered area.

770





**Table 2.** Mean values and uncertainties of sediment-entrapped $CH_4$ content, sediment-covered area, and sediment total porosity for Im Griess (IMG), Griessfirn (GRF), Griessen (GRI), Wildstrubel (WIL), and Tsanfleuron (TSA) glacier forefields located within the Helvetic Nappes of Switzerland. Also listed are individual nappes and major geological formations, from which glacier-forefield sediments are derived.

| Glacier forefield | Entrapped $CH_4$ content ($\mu$g $CH_4$ g$^{-1}$ d.w.) | Sediment-covered area (km$^2$) | Sediment porosity[a] (-) | Sediment origin[b] | |
| --- | --- | --- | --- | --- | --- |
| | $\overline{C}_{CH4} \pm \sigma_{\overline{C}_{CH4}}$ [c] | $\overline{A}_{sed} \pm \sigma_{\overline{A}_{sed}}$ | $\overline{\theta}_{t,sed} \pm \sigma_{\overline{\theta}_{t,sed}}$ | Nappes | Geological formations[d] |
| IMG | 6.51 ± 0.56 | 2.03 ± 0.42 | 0.44 ± 0.05 | Kammlistock | Quinten, Schrattenkalk, Stad, Zementstein |
| GRF | 5.59 ± 0.54 | 2.27 ± 0.40 | 0.42 ± 0.02 | Kammlistock Griessstock | Betlis, Helvetic Siliceous Limestone, Öhrli, Quinten, Zementstein |
| GRI | 7.03 ± 0.91 | 2.04 ± 0.55 | 0.38 ± 0.04 | Axen | Bommerstein, Hochstollen, Quinten |
| WIL | 39.41 ± 2.62 | 6.35 ± 1.01 | 0.43 ± 0.02 | Wildhorn Doldenhorn | Garschella, Öhrli, Quinten, Schilt, Schrattenkalk, Seewen, Tierwis |
| TSA | 33.74 ± 3.31 | 3.48 ± 0.91 | 0.44 ± 0.05 | Wildhorn Diablerets | Betlis, Helvetic Siliceous Limestone, Öhrli,Schrattenkalk, Tierwis, Tsanfleuron Member, Pierredar |

[a] adopted from Nauer et al. (2012).

[b] information obtained from the Geological Atlas of Switzerland 1:25.000 (online at maps.geo.admin.ch), Swiss Federal Office of Topography (swisstopo).

[c] standard error of the mean (SE).

[d] in alphabetical order.



**Table 3.** Data used for upscaling the mass of sediment-entrapped CH4 from five sampled glacier forefields (Im Griess (IMG), Griessfirn (GRF), Griessen (GRI), Wildstrubel (WIL), and Tsanfleuron (TSA)) to all calcareous glacier-forefields in Switzerland.

| Parameter | Value | Data source |
|---|---|---|
| Total Alpine area in Switzerland | $2.29 \times 10^4 \pm 504$ km$^2$ | Tectonic Map of Switzerland 1:500.000 (swisstopo) |
| Area of calcareous bedrock | $1.25 \times 10^4 \pm 275$ km$^2$ | Tectonic Map of Switzerland 1:500.000 (swisstopo) |
| Total glaciated area in 1850 | $1.62 \times 10^3 \pm 36$ km$^2$ | Zemp et al. (2008) |
| Total glaciated area in 2010 | $9.44 \times 10^2 \pm 21$ km$^2$ | Fischer et al. (2014) |
| Mean entrapped CH4 content, $\overline{C}_{CH4}$ | $18.5 \pm 4.4$ µg CH$_4$ g$^{-1}$ d.w. | Mean of averages for five glacier forefields |
| Mean sediment thickness, $\overline{T}_{sed}$ | $10.0 \pm 3.0$ m | Measurements in GRF glacier forefield |
| Mean sediment-particle density, $\overline{\rho}_{sed}$ | $2.71 \pm 0.15$ g cm$^{-3}$ | Daly (1935) |
| Mean total sediment porosity, $\overline{\theta}_{t,sed}$ | $0.44 \pm 0.05$ | Derived from data of Nauer et al. (2012) |
| Mass of sediment-entrapped CH4, $\overline{m}_{CH4}$ | $1.04 \times 10^5 \pm 3.7 \times 10^4$ t CH$_4$ | Eq. (1), this manuscript |