# Peer review of "Quantity and distribution of methane entrapped in sediments of calcareous, Alpine glacier forefields"

_Biogeosciences, 2019_

## Referee Comment (RC1) · David Archer (Referee) · 27 Mar 2020

This is an interesting, thorough characterization of the distribution of methane trapped within CaCO3 in glacial fore field deposits. The methane is released when the CaCO3 is dissolved in acid, a somewhat aggressive analog for chemical weathering. The authors are very careful not to overstate the implications of their data to the global methane cycle or climate, even though as they point out, the actual quantity of methane is rather high relative to other regional metrics. The primary motivation for investigating this is curiosity, which is a perfectly fine motivation for publication. I would be curious whether the methane has a measurable impact on the microbiology within the sediments; whether there is metabolic energy to be gained by reacting the methane with anything available, and whether RNA or proteomics of some other type of biotech char-

acterization could detect this activity. Probably any methane-driven metabolic activity would be at low level, given the apparently conservative behavior of the methane that the paper documents. This is not a suggestion for idea for the current paper, obviously, which I would recommend for publication as is, with only one editorial suggestion, from line 73, "virtually omnipresent" could be changed to "found virtually everywhere" or something like that. The former phrase makes the methane itself seem virtual.

---

## Referee Comment (RC2) · Anonymous Referee #2 · 28 Mar 2020

GENERAL COMMENTS: The topic of the reviewed manuscript (MS) is the occurrence spatial distribution and estimated total mass of methane (CH4) entrapped in calcareous sediment and bedrock in glacier forefields in the Swiss Alps. The topic is both novel and relevant for the improved understanding of terrestrial CH4 reservoirs and their potential source of emission to the atmposhere. The current study takes of where Zhu et al (2018) ends, with a clearly formulated aim (ll 87-89): "To extend the work of Zhu er at (2018) to other calcareous glacier forefields located in different regions of the Swiss Alps, and to assess the distribution of entrapped CH4 contents within and compare total mass of entrapped CH4 between all sampled glacier forefields". The study is well designed with clear descriptions of the work that has been done. However, since the study can be viewed as almost a part 2 of a larger study of entrapped CH4 in

calcareous sediment in Alpine catchments, with the Xhu et al (2018) paper as part 1, it could benefit from reflecting this more clearly. In particular, I would recommend that the manuscript is abbreviated significantly, to sharpen the focus, novelty and importance of the extended study, making it as short and concise as possible using referencing to the Zhu et al (2018) paper when it comes to more general background information. Finally, I recommend that the use of sediment age and landform as explanatory variables for the potential amount of entrapped CH4 present in the sediment is revised according to the specific comments.

In summary, I recommend publication of the study following a careful revision of the manuscript.

SPECIFIC COMMENTS: Ll 30-69: The two opening paragraphs are almost identical to the introduction of Zhu et al (2018) and does not mention subglacial/ glacial CH4. These two paragraphs could be shortening into a few referenced sentences with reference to Zhu et al (2018), in order for the MS to more quickly get to the essence of this study (from line 60 and onwards).

l.125. In this section the three steps of the fieldwork of stage I is described. However, the actual testing of the effect of sediment depth, sediment exposure age and landform on entrapped CH4 as well as total mass estimation comes after the fieldwork as it is based on data analysis. The text describing this should therefore be placed at a more appropriate stage in the MS.

l. 129. Is there a difference between a randomized design and a completely randomized design? If yes, please explain. If no, remove completely.

l. 152 What is the uncertainty on depth estimated based of the electrical resistivity tomography (ERT) method, and how does this uncertainty propagate into total CH4 mass best estimates? Section 2.2.1 in general. Can you absolutely rule out that no CH4 is chemically produced during the acid dissolution of the carbonate rock (by e.g. cold temperature version of similar processes as the high temperature

conversion of CO2 into CH4 using reduced metals as catalyst as decribed in e.g. https://onlinelibrary.wiley.com/doi/full/10.1002/cssc.200900152)

L 178. Good example of efficient referencing to previous literature on the same topic, saving space in this MS.

L. 179. Why where particles larger than 20 mm excluded?

L. 189. Maybe expand a little on how the initial tests were conducted in order to reach your methodological conclusion

L. 204. The number of samples used should be more clearly stated. Avoid using "About five samples..."

L.232. Title of 2.3.2 should include a description of what is being estimated, e.g. "Estimation of XXXX for five glacier forefields".

L. 237 and onwards. Natural chemical dissolution of carbonate rock from carbonic acid is an important weathering process over geological timescales and is also likely to be a process of relevance in the glacial forelands in which the study is performed. With this time perspective in mind, the reason for choosing the time difference between 1850 (asserted latest glacial maximum) and 2018 (same but minimum) as the boundary conditions for the CH4 upscaling appear somewhat arbitrary. I understand of course that it is an operational boundary condition for identifying the area of the current glacial forelands. But the temporal relevance of the phenomena that you investigate, where the natural chemical weathering rates are in fact the rate limiting factor for potential CH4 emissions needs to be justified on a bigger timescale. Especially the implied relationship the last glacial maximum could serve as a time zero for CH4 release (see also line 326) and that the current climatic development should in fact increase emission to the atmosphere, which may very well not be the case, if the overall rate limiting factor is not glacial coverage, but rather the kinetics of natural carbonate weathering, which can take place both in a warm-based subglacial environment as well as in the current

pro-glacial settings.

L. 273. Your conclusion that the gas must be of thermogenic origin and has not been altered by physical/chemical weathering appear correct. However, this particular conclusion supports the view that sediment age (i.e. time since last exposure) could be an irrelevant measure on the time scale that you operate, which also your statistical analysis show (l. 283).

L. 283. The term Landform is a less exact term for the variable than e.g. mineralogy of the sediment, parent material for the sediment or similar. As stated in the MS in seems that you argue that the landform itself has a significant effect on the potential CH4 content, while I believe you mean that significant difference in entrapped CH4 content is observed between different landforms. The sediment in the different landforms (floodplain, terrase, sand hill, etc) could potentially originate in contrasting parent material, and the variation in entrapped CH4 is more likely an effect of this, rather than the landform itself, the time of deposition or the time since the most recent exposure to the atmosphere after year 1850. I recommend that both this section and the MS in general is revised to reflect this relationship.

L.302 Similar to the comment above, the distance between landform and its relationship to entrapped CH4 is more likely to be a proxy of the parent material of the sediment and deposition history than the time elapsed since the areas were covered by glaciers, i.e. glacier extend and sediment age as defined could be irrelevant properties for explaining the inferred amount of entrapped CH4.

L. 319 Please use a quantative terms, rather than the qualitative term "little".

L.326 as stated in the comment above, more arguments should be provided to back up the assumption that maximum glaciated area in year approx. 1850 is the original value as stated here. It seems that, the original value could more correctly be described as the start value for your estimate of the exposed proglacial area following the most recent major glacial retreat, which could or could not have a direct influence

of the amount of CH4 stored in the sediment. However, the relevant age of the sediments CH4 content (i.e. the time the CH4 was trapped in the rock which by weathering became sediment) is not related to the point in time in which it most recently was exposed to the atmosphere (i.e. not covered by ice), nor to the point in time where the sediment was deposited in the current landform. It is highly likely that the sediment that you sample here and now, on several other points in tie have been exposed to the atmosphere without either incorporating more CH4 or releasing parts of the currently entrapped CH4.

L.329. Consider the wording in the sentence: " From these numbers, the total mass of sediment-entrapped CH4 in all Swiss glacier forefields derived from calcareous bedrock was computed...." I recommend that this sentence should be revised to more appropriately reflect what the study has done, namely to give a best estimate of the entrapped CH4 in an area of the glacier forefields corresponding to the area extending from the current position of the glaciers to their reconstructed position during the most recent glacial maximum, which is different from "all Swiss glacier forefields". The findings and associated increase in scientific understanding is sufficiently strong, novel and interesting in itself, and I see no reason for trying to upscale the potential amount of CH4, which at the current level of understanding will be very uncertain.

L. 376 Little variation in entrapped CH4 across sediment depth and exposure age, indicate that the CH4 concentration is not dependent on recent transformations or release, but an inherent property reflecting the CH4 content of the parent material (as also indicated in L. 383). Again, sediment age as defined does not seem to be a very relevant explanatory variable.

L.386/387. What "major alterations" do you suggest that the sediment has undergone during and after the erosion of the parent materiel? Usually, physical erosion of bedrock primarily reduces the grain size of the material in question without any further alterations to the matrix of the grain (unless the material undergoes diagenesis). I believe your observations point towards the opposite, namely that the sediment has

not undergone any significant alterations with respect to entrapped CH4 and that this property is indeed one of the key take-home messages of your story, i.e. that large quantities of entrapped CH4 is present, but not very likely to be quickly mobilized by natural weathering with following release to the atmosphere. The importance of this is of course linked to recent discoveries of subglacial CH4 emissions (as included in your references), in which the sediment entrapped CH4 is likely not a major contributor, unless there is significant subglacial dissolution of calcareous material with entrapped CH4.

L- 404/405 and 413/414. Yes, differences in CH4 content of the parent rock is likely the main explanation for the observed variability. An improved mineralogical investigation of the sediment in the various landforms would be able to test whether the sediment in the floodplain is significantly different that the other two landform, thereby providing a possible explanation for the observed differences in CH4 content.

L 426/427 What is the relevance of comparing a large, entrapped, immobilized and thermogenic CH4 volume in sediment with no proven interaction with the atmosphere to a mobile, biogenic CH4 pool in lakes, wetland and wild animals? These are two completely different carbon cycles, with very contrasting element cycling times.

L 433 - 441. Why is it important to narrow down the uncertainty of how much CH4 is indeed present in the entire area going beyond your study area, if the CH4 is not mobile?

L441 – L 453. CH4 emission release rates by chemical weathering is likely to be orders of magnitude lower than reported rates of microbial CH4 oxidation in soil and sediment. The described scenario is quite hypothetical and non-documented. To strengthen the scope of the study and highlight its importance, I recommend to remove this last section of the MS dealing with microbial oxidation and exchange with the atmosphere, as this is most likely not happening at a rate with any significance for biological CH4 turnover.

L. 455-468. Very good summary of the presented work and conclusions.

L. 469-474 Somewhat speculative when the data suggest the opposite, i.e. that CH4 is very stable within sediment and not released due to weathering at any significant rate (no significant difference with sediment depth+ entrapped CH4 in sediment reflect that of parent material). I suggest removing this part of the conclusion to make your story more focused, and not end on a speculatory note, when you in fact have quite strong and novel data.

Figure 2: Good idea to show sample point and profiles on a map. However As mentioned above, the concept of sediment age as an explanatory variable does not seem justified.

Figure 3: The difference between minimum and maximum glacial extend could be irrelevant as an explanatory variable.

Figure 7: More info on mineralogy and parent material would be useful to better characterize and understand the shown differences in entrapped CH4. The absolute unit "Mass of entrapped CH4 (t)" is very dependent on your upscaling and its associated uncertainty. I suggest to revise figure 7a, the show entrapped CH4 in relative terms (could be g CH4 per ton sediment or similar) to better the variation span in entrapped CH4 per sediment type (i.e. what you call landform).

Table 2: Sediment porosity: It is not clear if the the parameter "porosity" indicated intra-grain porosity (i.e. amount of pore volume within the sediment) or intra-grain porosity (i.e. amount of pore volume between grains, which must be assumed to be occupied by atmospheric air at approximately 1.9 ppm CH4). Please clarify.

---

## Author Comment (AC1) · 6 May 2020

David Archer (Referee) d-archer@uchicago.edu

This is an interesting, thorough characterization of the distribution of methane trapped within CaCO3 in glacial fore field deposits. The methane is released when the CaCO3 is dissolved in acid, a somewhat aggressive analog for chemical weathering. The authors are very careful not to overstate the implications of their data to the global methane cycle or climate, even though as they point out, the actual quantity of methane is rather high relative to other regional metrics. The primary motivation for investigating this is curiosity, which is a perfectly fine motivation for publication. I would be curious whether the methane has a measurable impact on the microbiology within the sediments; whether there is metabolic energy to be gained by reacting the methane with anything available, and whether RNA or proteomics of some other type of biotech characterization could detect this activity. Probably any methane-driven metabolic activity would be at low level, given the apparently conservative behavior of the methane that the paper documents. This is not a suggestion for idea for the current paper, obviously, which I would recommend for publication as is, with only one editorial suggestion, from line 73, "virtually omnipresent" could be changed to "found virtually everywhere" or something like that. The former phrase makes the methane itself seem virtual.

We would like to thank the Reviewer for the overall positive assessment of our manuscript.

We agree with the Reviewer that it would be highly interesting to investigate whether this entrapped $CH_4$ has an impact on the microbiology within the sediments, particularly on methane-oxidizing bacteria (MOB). In earlier studies (e.g., Chiri et al. 2017) we have confirmed the presence and activity of aerobic MOB utilizing atmospheric $CH_4$ in calcareous glacier-forefield sediments. This group of MOB is well adapted to utilizing $CH_4$ at low levels (< 2 $\mu$L $L^{-1}$ in gas phase) in this environment. Indeed, a set of recently conducted experiments in our laboratory yielded first indications that these MOB may utilize (i.e., oxidize) trace amounts of previously sediment-entrapped $CH_4$, which "leaks" from the calcareous sediments into the sediment-gas phase. However, we consider these experiments to be preliminary, additional experiments will have to be conducted for confirmation. But as indicated by the Reviewer, this topic is beyond the scope of the current manuscript.

We also agree with the Reviewer's editorial comment, and have re-phrased the sentence in question to now read:

"…we established that entrapped $CH_4$ was present in nearly all sediment and bedrock samples collected throughout this catchment…"

---

## Author Comment (AC2) · 6 May 2020

GENERAL COMMENTS:
The topic of the reviewed manuscript (MS) is the occurrence spatial distribution and estimated total mass of methane ($CH_4$) entrapped in calcareous sediment and bedrock in glacier forefields in the Swiss Alps. The topic is both novel and relevant for the improved understanding of terrestrial $CH_4$ reservoirs and their potential source of emission to the atmposhere. The current study takes of where Zhu et al (2018) ends, with a clearly formulated aim (ll 87-89): "To extend the work of Zhu er at (2018) to other calcareous glacier forefields located in different regions of the Swiss Alps, and to assess the distribution of entrapped $CH_4$ contents within and compare total mass of entrapped $CH_4$ between all sampled glacier forefields". The study is well designed with clear descriptions of the work that has been done. However, since the study can be viewed as almost a part 2 of a larger study of entrapped $CH_4$ in calcareous sediment in Alpine catchments, with the Xhu et al (2018) paper as part 1, it could benefit from reflecting this more clearly. In particular, I would recommend that the manuscript is abbreviated significantly, to sharpen the focus, novelty and importance of the extended study, making it as short and concise as possible using referencing to the Zhu et al (2018) paper when it comes to more general background information. Finally, I recommend that the use of sediment age and landform as explanatory variables for the potential amount of entrapped $CH_4$ present in the sediment is revised according to the specific comments. In summary, I recommend publication of the study following a careful revision of the manuscript.

We would like to thank the Reviewer for a detailed review, and for the overall positive assessment of our manuscript. In the General Comments section, the Reviewer makes two recommendations, (i) to shorten the manuscript and (ii) to revise the use of sediment age and landform as explanatory variables. As both issues are also raised under Specific Comments, we will address these issues in the Specific Comments section below.

SPECIFIC COMMENTS:
1.) Ll 30-69: The two opening paragraphs are almost identical to the introduction of Zhu et al (2018) and does not mention subglacial/ glacial $CH_4$. These two paragraphs could be shortening into a few referenced sentences with reference to Zhu et al (2018), in order for the MS to more quickly get to the essence of this study (from line 60 and onwards).

We agree with the reviewer that the first two paragraphs of the Introduction section can be shortened, and we shortened them from 29 to 18 lines of text. On the other hand, we strongly feel that core topics relevant to this manuscript such as the distinction between microbial, thermogenic, and abiotic $CH_4$ should be presented to readers in the Introduction section rather than just referring to our previous publication.
In addition, we feel that our introduction into the topic of subglacial $CH_4$ (third paragraph) may also be shortened, as subglacial $CH_4$ is not the topic of this manuscript. Moreover, we feel that the reviewer was confused with our use of the term "sediment age" (see also discussion of comments below), in that we use it to indicate time since deglaciation rather than "absolute" sediment age (time since the formation of sediments). We feel that a clarification regarding this term is needed prior to its (previously first) appearance in our

objective statement. We previously provided such a statement in our Methods section, whereas we now introduce and provide a clarification of this term at the end of the third paragraph of our Introduction. The clarification reads:

"In this context, sediment age refers to the number of years sediment has been exposed to the atmosphere following glacier retreat. Note that both terms, sediment age and landform, serve as proxies for all edaphic variations present in these sediments at different locations within the glacier forefield. We will adopt this convention and use the terms sediment age and landform in this fashion throughout this paper." (l. 59)

2.) l.125. In this section the three steps of the fieldwork of stage I is described. However, the actual testing of the effect of sediment depth, sediment exposure age and landform on entrapped CH4 as well as total mass estimation comes after the fieldwork as it is based on data analysis. The text describing this should therefore be placed at a more appropriate stage in the MS.

There appears to be a misunderstanding. The Reviewer suggests that we conducted all stage 1 field sampling prior to analyses in the laboratory, and therefore the order of description in our method section should be changed. But this is incorrect. We conducted the work exactly as described in our Methods section 2.1.1. In other words, after collecting samples to test for sediment depth, these samples were analysed in the laboratory for $CH_4$ content. The results of this analysis was then used to adapt the sampling scheme of the subsequent step, in which we tested for the effects of sediment age and landform. This is mentioned in both the Methods section ("The sampling depth of 20 cm below ground surface was chosen based on our results from the previous step"; l. 145) and the Results section ("Thus, we subsequently proceeded to collect sediments from 20 cm depth only, and assumed these samples to be representative in terms of entrapped $CH_4$ content for the entire sediment thickness."; l. 279).
To further clarify this issue, we added the following sentence to the end of the Methods paragraph describing step 1: "The result of this analysis was then used to adapt the sampling scheme for the following step." (l. 134)

3.) l. 129. Is there a difference between a randomized design and a completely randomized design? If yes, please explain. If no, remove completely.

We agree with the Reviewer. As there is no difference, we deleted the word "completely".

4.) l. 152 What is the uncertainty on depth estimated based of the electrical resistivity tomography (ERT) method, and how does this uncertainty propagate into total CH4 mass best estimates? Section 2.2.1 in general. Can you absolutely rule out that no CH4 is chemically produced during the acid dissolution of the carbonate rock (by e.g. cold temperature version of similar processes as the high temperature conversion of CO2 into CH4 using reduced metals as catalyst as decribed in e.g. https://onlinelibrary.wiley.com/doi/full/10.1002/cssc.200900152)

The uncertainty of ERT on the estimate of mean sediment depth is documented in Table 1. It amounts to up to 50% of the respective mean depth value. This large uncertainty is dominated by the high variability in sediment depth encountered within the glacier forefield, rather than by measurement uncertainties in the field and uncertainties introduced during the inversion of apparent resistivities (typically <10% (Loke, 2000)). The propagation of this uncertainty into total $CH_4$ mass estimation is mathematically described in Eq. 2, and its contribution to the total uncertainty in estimated $CH_4$ mass is shown in Figure 7b. To better indicate the origin of the uncertainty in sediment depth, we modified a sentence in the results section to now read:

"Uncertainties in individual $m_{CH4}$ values up to ~50% mostly arose from uncertainties in $T_{sed}$ (dominated by the large variability in sediment depth across the GRF sampling zone) and to a smaller degree from uncertainties in $C_{CH4}$." (l. 300)

We are familiar with the process of "methanization" as described in the reference provided by the Reviewer, but not with a "cold-temperature version" under the prevailing experimental conditions. As noted by the Reviewer, methanization takes place at high temperatures (above 300°C) and/or pressures in the presence of a catalyst and in a $H_2$ atmosphere. None of these conditions are met during our acidification experiments. Apart from conducting these experiments at ambient temperature (21°C), we are certain that there is no $H_2$ (which would serve as the electron donor in the chemical reduction of $CO_2$ during methanization) present in our assays, as the headspace of all samples was flushed with $N_2$ gas prior to acidification treatment (l. 182).
But more importantly, we have previously provided unequivocal evidence that $CH_4$ release from these calcareous sediments can be induced by other means than acidification (Nauer et al., 2014). In that paper we demonstrated that similar amounts of $CH_4$ are released by mechanical disturbance (hammering) in the field, and that a substantial quantity of $CH_4$ is released in the laboratory by the addition of water and subsequent sonification treatment. None of these treatments involve the use of acid.

5.) L 178. Good example of efficient referencing to previous literature on the same topic, saving space in this MS.

We thank the Reviewer for this assessment.

6.) L. 179. Why where particles larger than 20 mm excluded?

We adopted this threshold in particle size from our previous work (Zhu et al., 2018) for compatibility, but also for the following reason: On the international (ISO) scale for soil classification, 20 mm marks the boundary between medium and coarse gravel. Apart from the topmost sediment layer (< 5 cm depth, not sampled), particles > 20 mm in size were rare in our samples, thus did not appear representative of the bulk sediment in these sediments.

7.) L. 189. Maybe expand a little on how the initial tests were conducted in order to reach your methodological conclusion

We agree with the Reviewer that some more information on this issue would be helpful to readers. In these initial tests we compared hammering with sawing. Using both methods we collected rock fragments both from the surface and from the core of larger rocks. Obtaining fragments from the core required substantially more hammering and more sawing, thus a much longer duration of the mechanical treatment. But subsequent analyses of these rock fragments showed that differences in geochemical parameters were insignificant. This we took as evidence that there was no adverse effect of sawing or hammering on the fragment's geochemical parameters. To provide more information in concise fashion to readers, we replaced the sentence in question with the following statement:

"Initial tests in which we compared hammering with sawing to obtain rock fragments both from the surface and from the core of larger rocks showed insignificant effects on the fragments' entrapped $CH_4$ contents and other geochemical parameters. As the duration of the respective mechanical treatment varied greatly between the collected fragments, we consider this as evidence that neither hammering nor sawing had an adverse effect on measured geochemical parameters." (l. 190)

8.) L. 204. The number of samples used should be more clearly stated. Avoid using "About five samples. . ."

We fully agree with the Reviewer and have rephrased the sentence in question to read:

"A total of 31 sediment and bedrock samples from the five glacier forefields were selected for stable carbon-isotope analysis of entrapped $CH_4$ ($\delta^{13}C_{CH4}$)." (l. 208)

9.) L.232. Title of 2.3.2 should include a description of what is being estimated, e.g. "Estimation of XXXX for five glacier forefields".

We agree with the Reviewer. However, for consistency we have included the description also in titles of sections 2.3.1 and 2.3.3. Title 2.3.2 now reads:

"Estimation of entrapped $CH_4$ mass for the five glacier forefields (IMG, GRF, GRI, WIL, TSA)" (l. 235)

10.) L. 237 and onwards. Natural chemical dissolution of carbonate rock from carbonic acid is an important weathering process over geological timescales and is also likely to be a process of relevance in the glacial forelands in which the study is performed. With this time perspective in mind, the reason for choosing the time difference between 1850 (asserted latest glacial maximum) and 2018 (same but minimum) as the boundary conditions for the CH4 upscaling appear somewhat arbitrary. I understand of course that it is an operational boundary condition for identifying the area of the current glacial forelands. But the temporal relevance of the phenomena that you investigate, where the natural chemical weathering rates are in fact the rate limiting factor for potential CH4 emissions needs to be justified on a bigger timescale. Especially the implied relationship the last glacial maximum could serve as a time zero for CH4 release (see also line 326) and that the current climatic development should in fact increase emission to the atmosphere, which may very well not be the case, if the overall rate limiting factor is not glacial coverage, but rather the kinetics of natural carbonate weathering, which can take place both in a warm-based subglacial environment as well as in the current pro-glacial settings.

We agree with the Reviewer that chemical weathering is indeed a slow process taking place over geologic timescales. However, with the transition from subglacial to proglacial sediment as a result of glacier retreat, forefield sediments undergo changes other than mere chemical weathering. As soil formation is initiated following deglaciation, microbial and pioneer plant colonisation may alter sediment characteristics, which are visually noticeable within a few decades following deglaciation, and certainly within a 100 to 150 year chronosequence. These changes may include chemical characteristics including TOC, nutrient levels, and pH (Stevens & Walker, 1970; Bernasconi et al., 2008; Lazzaro et al., 2009; Chiri et al., 2015). We strongly feel that it is a fair and worthy question to ask whether sediment age (used here as time since deglaciation and as a proxy for edaphic variations) affects sediment-entrapped $CH_4$ contents or not.

As hinted by the Reviewer, we indeed use the time interval from the end of the little ice age until present time _exclusively_ to estimate the surface area of the glacier forefields, because the 1850 end moraine is often a visible feature in the landscape, defining the extent of the glacial advance at the end of the little ice age. However, we carefully checked our manuscript to assure that nowhere we imply a relationship that the last glacial maximum could serve as a time zero for $CH_4$ release, nor that current climatic development should increase emissions to the atmosphere.

11.) L. 273. Your conclusion that the gas must be of thermogenic origin and has not been altered by physical/chemical weathering appear correct. However, this particular conclusion supports the view that sediment age (i.e. time since last exposure) could be an irrelevant measure on the time scale that you operate, which also your statistical analysis show (l. 283).

We agree with the Reviewer. This is indeed an important result of this study, and it is in agreement with our finding that effects of sediment age on $CH_4$ contents were insignificant. The implications of these findings (entrapped $CH_4$ appears stable in its entrapped state) are discussed beginning on line 383. We see no inconsistency here.

12.) L. 283. The term Landform is a less exact term for the variable than e.g. mineralogy of the sediment, parent material for the sediment or similar. As stated in the MS in seems that you argue that the landform itself has a significant effect on the potential CH4 content, while I believe you mean that significant difference in entrapped CH4 content is observed between different landforms. The sediment in the different landforms (floodplain, terrase, sand hill, etc) could potentially originate in contrasting parent material, and the variation in entrapped CH4 is more likely an effect of this, rather than the landform itself, the time of deposition or the time since the most recent exposure to the atmosphere after year 1850. I recommend that both this section and the MS in general is revised to reflect this relationship.

We agree with the Reviewer that this issue requires clarification. We used "landform" as one variable to spatially discretize the glacier forefield sampling zone. The other variable we used is sediment age (time since deglaciation, see answer to comment 1). In doing so, we follow a previous spatial discretization of the forefield sampling zone (Chiri et al., 2017). Moreover, sediment age and landform is terminology commonly used in research papers dealing with studies in glacier forefields. For compatibility reasons we prefer to keep using these terms for the spatial discretization. However, we fully agree with the Reviewer that "landform" is not a causal variable, i.e., it is not the cause for the different sediment-entrapped $CH_4$ contents detected. Rather, and as is the case also for sediment age, both terms serve as proxies for all edaphic variations present in sediments at these locations in the glacier forefield. Therefore, we strongly feel that an up-front clarification on the use of these terms should be provided already in the Introduction (before they are used in the objective statement). This is why we now introduce these terms in our Introduction, and provide the following clarification:

"In this context, sediment age refers to the number of years sediment has been exposed to the atmosphere following glacier retreat. Note that both terms, sediment age and landform, serve as proxies for all edaphic variations present in these sediments at different locations within the glacier forefield. We will adopt this convention and use the terms sediment age and landform in this fashion throughout this paper." (l. 59)

Together with our (modified) statement in the Method section, we feel this provides adequate clarification:

"During stage I in summer 2016, we performed a detailed investigation on the spatial distribution of sediment-entrapped $CH_4$ within a designated sampling zone at the GRF glacier forefield, using high spatial-resolution sampling to determine variations in entrapped $CH_4$ contents in relation to sediment depth, sediment age, and glacier-forefield landforms. The GRF forefield was chosen for this purpose mainly because it features well-defined sediment-age classes and well-developed, clearly distinguishable landforms within a previously discretized and characterized sampling zone (Chiri et al., 2015; Chiri et al., 2017)." (l. 119)

To remind the reader on the use of these terms, we also modified the first sentence of the Results paragraph in question to read:

"The effects of the proxies sediment age and landform on entrapped $CH_4$ contents were tested using…". (l. 280)

Finally, we fully agree with the Reviewer that observed differences in in entrapped $CH_4$ contents in different landforms may be related to differences in parent bedrock from which these sediments are derived. We modified a paragraph in our Discussion section to better address this issue:

"In contrast to sediment depth and sediment age, we detected a small but significant difference in mean sediment-entrapped $CH_4$ content between landforms within the GRF sampling zone. Specifically, mean entrapped $CH_4$ content in floodplain sediments was significantly higher than in terrace and sandhill sediments (Table 1). We can only speculate about possible reasons for this observation. One reason could be that floodplain sediments, intermittently removed and deposited by the glacial stream during and after flooding events, originate from locations outside of our sampling zone, i.e., from different parent bedrock (Fig. S2). It is therefore possible that we missed to sample parent bedrock types (e.g., from steep rock walls) with different (in this case higher) entrapped $CH_4$ contents in this or any of the other glacial catchments." (l. 398)

13.) L.302 Similar to the comment above, the distance between landform and its relationship to entrapped CH4 is more likely to be a proxy of the parent material of the sediment and deposition history than the time elapsed since the areas were covered by glaciers, i.e. glacier extend and sediment age as defined could be irrelevant properties for explaining the inferred amount of entrapped CH4.

We agree with the Reviewer that distance is not a causal variable. But in our results section we merely state our finding with due caution in that we say "…with distance between the forefields playing an *apparently* important role." In the Discussion (l. 405), we then make the link to the parent material, i.e. the differences in lithology and tectonic settings. We have slightly the central statement to now read:

"This may be explained by the fact that sediments in glacier forefields located in close proximity to one another are, at least in part, derived from the same individual nappes and geological formations contained therein." (l. 406) … "Hence, this result supports our previous hypothesis that differences in lithology, mineralogy, and tectonic settings between individual nappes play an important role in determining bedrock- and thus sediment-entrapped $CH_4$ contents…". (l. 409)

14.) L. 319 Please use a quantative terms, rather than the qualitative term "little".

We agree with the Reviewer. The sentence in question now reads:

"Conversely, entrapped $CH_4$ contents, sediment porosity, and sediment-particle density together contributed ≤ 16% to the calculated uncertainties." (l. 315)

15.) L.326 as stated in the comment above, more arguments should be provided to back up the assumption that maximum glaciated area in year approx. 1850 is the original value as stated here. It seems that, the original value could more correctly be de- scribed as the start value for your estimate of the exposed proglacial area following the most recent major glacial retreat, which could or could not have a direct influence of the amount of CH4 stored in the sediment. However, the relevant age of the sediments CH4 content (i.e. the time the

CH4 was trapped in the rock which by weathering became sediment) is not related to the point in time in which it most recently was ex- posed to the atmosphere (i.e. not covered by ice), nor to the point in time where the sediment was deposited in the current landform. It is highly likely that the sediment that you sample here and now, on several other points in tie have been exposed to the atmosphere without either incorporating more CH4 or releasing parts of the currently entrapped CH4.

This comment is similar to comment 10 and we refer to our answer to that comment. We strongly feel that there was a misunderstanding on our use of the 1850 maximum glaciated area. As stated above, the *only* purpose of using the 1850 glacial extent together with the extent of 2010 was to estimate the glacier-forefield area that has been exposed as a result of deglaciation since 1850. In that sense we follow the commonly used definition of a glacier forefield ("area between the moraines of modern (or post-glacial) advances (e.g. greatest extent as around 1850/60) and today's glacier outlines"; Glacier Monitoring of Switzerland (GLAMOS)). Please note that nowhere in the paragraph in question nor in the entire manuscript do we claim that 1850 marks an original value or a time zero for entrapped CH4 contents.

16.) L.329. Consider the wording in the sentence: " From these numbers, the total mass of sediment-entrapped CH4 in all Swiss glacier forefields derived from calcareous bedrock was computed. . .." I recommend that this sentence should be revised to more appropriately reflect what the study has done, namely to give a best estimate of the entrapped CH4 in an area of the glacier forefields corresponding to the area extending from the current position of the glaciers to their reconstructed position during the most recent glacial maximum, which is different from "all Swiss glacier forefields". The findings and associated increase in scientific understanding is sufficiently strong, novel and interesting in itself, and I see no reason for trying to upscale the potential amount of CH4, which at the current level of understanding will be very uncertain.

The entire Results paragraph in question deals with the upscaling of the results we obtained in five glacier forefields to all calcareous glacier forefields in Switzerland. We agree with the Reviewer that the estimate from this upscaling must be associated with a large uncertainty. Nonetheless, we feel that providing a first albeit rough estimate is a worthy objective, particularly when an estimate of the associated uncertainty can be provided, which is what we have done. As suggested by the Reviewer, and to indicate that this is not a precise computation but just an estimate, we modified the last sentence of this paragraph to now read:

"From these numbers, we estimated the total mass of sediment-entrapped CH4 in all Swiss glacier forefields derived from calcareous bedrock to $1.04 \times 10^5 \pm 3.7 \times 10^4$ t $CH_4$." (l. 325)

We note that in the discussion of these results, we intentionally state that we consider this a rough estimate only, repeating the substantial uncertainty involved. "Our first, rough estimate for the total quantity of $CH_4$ entrapped in sediments of all calcareous Swiss glacier forefields combined yielded a substantial mass of $1.04 \times 10^5 \pm 3.7 \times 10^4$ t $CH_4$, contained within a solid volume of ~2.1 $km^3$ glacier-forefield sediments." (l. 421)

17.) L. 376 Little variation in entrapped CH4 across sediment depth and exposure age, indicate that the CH4 concentration is not dependent on recent transformations or release, but an inherent property reflecting the CH4 content of the parent material (as also indi- cated in L. 383). Again, sediment age as defined does not seem to be a very relevant explanatory variable.

This comment relates back to previous comments on sediment age. Indeed, sediment age even used in the sense of a proxy (see comments 1, 10, 11 above) showed no significant effect on entrapped $CH_4$ contents. This is one result of this study, which is here discussed.

18.) L.386/387. What "major alterations" do you suggest that the sediment has under- gone during and after the erosion of the parent materiel? Usually, physical erosion of bedrock primarily reduces the grain size of the material in question without any further alterations to the matrix of the grain (unless the material undergoes diagenesis). I believe your observations point towards the opposite, namely that the sediment has not undergone any significant alterations with respect to entrapped CH4 and that this property is indeed one of the key take-home messages of your story, i.e. that large quantities of entrapped CH4 is present, but not very likely to be quickly mobilized by natural weathering with following release to the atmosphere. The importance of this is of course linked to recent discoveries of subglacial CH4 emissions (as included in your references), in which the sediment entrapped CH4 is likely not a major contributor, unless there is significant subglacial dissolution of calcareous material with entrapped CH4.

We agree with the Reviewer that the sentence in question is misleading, as we argue in this paragraph that $CH_4$ in forefield sediments appears relatively stable in its entrapped state. To avoid misinterpretation, we rephrased the sentence in question to read:

"Thus, although sediments have undergone erosion from the parent bedrock and subsequent weathering, changes in entrapped $CH_4$ geochemical characteristics appeared negligible." (l. 383)

The relevance to subglacial $CH_4$ emissions is less clear in our opinion and requires further study. In particular, several studies indicate that $CH_4$ in these systems is derived from microbial $CH_4$ production rather than being of thermogenic origin.

19.) L- 404/405 and 413/414. Yes, differences in CH4 content of the parent rock is likely the main explanation for the observed variability. An improved mineralogical investigation of the sediment in the various landforms would be able to test whether the sediment in the floodplain is significantly different that the other two landform, thereby providing a possible explanation for the observed differences in CH4 content.

We agree with the Reviewer. To indicate the value of such a mineralogical investigation to readers we have added a sentence to the end of the first paragraph in question to read:

"An improved mineralogical investigation of sediments in the various landforms would aid in clarifying this issue." (l. 405)

20.) L 426/427 What is the relevance of comparing a large, entrapped, immobilized and thermogenic CH4 volume in sediment with no proven interaction with the atmosphere to a mobile, biogenic CH4 pool in lakes, wetland and wild animals? These are two completely different carbon cycles, with very contrasting element cycling times.

Here we wanted to provide a comparison of our estimate of entrapped $CH_4$ mass to other $CH_4$ inventory data available for Switzerland. However, most inventory data are available in the form of $CH_4$ fluxes. We agree with the Reviewer that it is debatable how relevant such comparisons are, but we feel it provides readers with a sense for the magnitude of the $CH_4$ pool in glacier forefields. As a compromise, we have removed the numbers on $CH_4$ flux from a Swiss lake, and only mention the estimate of total $CH_4$ flux from natural and semi-natural sources in Switzerland.

"At first glance, this number appears large when compared with an estimate of annual $CH_4$ emissions to the atmosphere ($5.7 \times 10^3$ t $CH_4$) from natural and semi-natural sources in Switzerland." (l. 424)

21.) L 433 - 441. Why is it important to narrow down the uncertainty of how much CH4 is indeed present in the entire area going beyond your study area, if the CH4 is not mobile?

The reason for upscaling is described in the answer to comment 16, and we feel that reducing the uncertainty in our currently rough estimate is a worthy cause by itself. Moreover, at this time there is no experimental verification that entrapped $CH_4$ is truly immobile. On the contrary, a set of recently conducted experiments in our laboratory yielded first indications that trace amounts of sediment-entrapped $CH_4$ may slowly "leak" from the calcareous sediments into the sediment-gas phase (see our response to the comment of Reviewer 1). But these experiments are beyond the scope of this manuscript, and we consider them preliminary only, requiring experimental confirmation.

22.) L441 – L 453. CH4 emission release rates by chemical weathering is likely to be orders of magnitude lower than reported rates of microbial CH4 oxidation in soil and sediment. The described scenario is quite hypothetical and non-documented. To strengthen the scope of the study and highlight its importance, I recommend to remove this last section of the MS dealing with microbial oxidation and exchange with the atmosphere, as this is most likely not happening at a rate with any significance for biological CH4 turnover.

We feel that a speculative statement at the end of the Discussion section is permissible, and that it does not weaken the manuscript, particularly in light of the unresolved question if traces of $CH_4$ may leak from these sediments (see previous comment). Moreover, we feel that we were careful in phrasing this speculation. We note that bacteria mediating atmospheric $CH_4$ oxidation make indeed a living on traces of $CH_4$ in the sub-ppm range. On the other hand, we fully agree with the Reviewer that such speculation should be avoided in the Summary and Conclusions section (see comment below).

23.) L. 455-468. Very good summary of the presented work and conclusions.

We thank the Reviewer for this assessment.

24.) L. 469-474 Somewhat speculative when the data suggest the opposite, i.e. that CH4 is very stable within sediment and not released due to weathering at any significant rate (no significant difference with sediment depth+ entrapped CH4 in sediment reflect that of parent material). I suggest removing this part of the conclusion to make your story more focused, and not end on a speculatory note, when you in fact have quite strong and novel data.

We agree with the Reviewer that the manuscript should not end with a speculation. We have therefore removed the paragraph in question.

25.) Figure 2: Good idea to show sample point and profiles on a map. However As mentioned above, the concept of sediment age as an explanatory variable does not seem justified.

This issue was addressed in comments 1 and 12. To remind the reader on the use of these terms as proxies, we also modified the caption of Fig. 2 to read:

"Sampling zone at Griessfirn (GRF) glacier forefield showing (a) blocks and sampling locations to study the effect of proxies sediment age and glacier-forefield landforms on entrapped $CH_4$ contents,…". (l. 703)

26.) Figure 3: The difference between minimum and maximum glacial extend could be irrelevant as an explanatory variable.

This issue was addressed in comments 10 and 15. The difference between minimum and maximum glacial extent was never intended as an explanatory variable, but was only used to estimate glacier-forefield area.

27.) Figure 7: More info on mineralogy and parent material would be useful to better char-acterize and understand the shown differences in entrapped CH4. The absolute unit "Mass of entrapped CH4 (t)" is very dependent on your upscaling and its associated uncertainty. I suggest to revise figure 7a, the show entrapped CH4 in relative terms (could be g CH4 per ton sediment or similar) to better the variation span in entrapped CH4 per sediment type (i.e. what you call landform).

We agree with the Reviewer that data on $CH_4$ contents is important to show for the five glacier forefields. This information is shown in Table 2. But as the five glacier forefields not only differ in sediment-entrapped $CH_4$ contents, but e.g. also in spatial extent (area), we feel that the determination of $CH_4$ mass and associated uncertainties for each of the forefields provides important, additional information and insight. We also argue that there is no upscaling involved in the data presented in Fig. 7a. For each of the five glacier forefields, we computed entrapped $CH_4$ mass based on measurements of $CH_4$ contents, and estimates of sediment thickness and sediment-covered area. (Upscaling, on the other hand, was used to provide a first estimate of entrapped $CH_4$ mass contained in all calcareous glacier forefield sediments in Switzerland).

28.) Table 2: Sediment porosity: It is not clear if the the parameter "porosity" indicated intragrain porosity (i.e. amount of pore volume within the sediment) or intra-grain porosity (i.e. amount of pore volume between grains, which must be assumed to be occupied by atmospheric air at approximately 1.9 ppm CH4). Please clarify.

We agree with the Reviewer that this needs clarification. We provide this clarification at the first instance where the term is introduced in the manuscript:

"…and $\theta_{t,sed}$ is total inter-particle sediment porosity, hereafter referred to as sediment porosity." (l. 211)

and also in the caption of Table 2:

"Mean values and uncertainties of sediment-entrapped $CH_4$ content, sediment-covered area, and total inter-particle sediment porosity …" (l. 755)

---

## Author Response (AR2)

**Eidgenössische Technische Hochschule Zürich**
**Swiss Federal Institute of Technology Zurich**

Prof. Dr. Martin H. Schroth
Institute for Biogeochemistry and
Pollutant Dynamics
Environmental Microbiology
Universitätstr. 16, CHN G50.2
CH-8092 Zürich
Switzerland
Phone: (+41) 44 – 633 6039
Fax:    (+41) 44 – 633 1122
martin.schroth@env.ethz.ch

Zürich, June 17, 2020

**To:**
**Editorial Office**
**Biogeosciences**

Dear Editor, dear Tina,

My co-authors and I would like to thank you for accepting our manuscript "Quantity and distribution of methane entrapped in sediments of calcareous, Alpine glacier forefields" for publication in ***Biogeosciences***.

We agree with your comment and have modified the Introduction section accordingly. It now reads (modifications highlighted in red):

[revised manuscript text omitted]